# Investigating and Mitigating Catastrophic Forgetting in Medical Knowledge Injection through Internal Knowledge Augmentation Learning

**Yuxuan Zhou**[1]**, Xien Liu**[*1]**, Xiao Zhang**[1]**, Chen Ning**[1]**, Shijin Wang**[2]**, Guoping Hu**[2]**, Ji Wu**[1,3,4]

[1]Department of Electronic Engineering, Tsinghua University, Beijing, China
[2]iFLYTEK Research, Hefei, China     [3]College of AI, Tsinghua University, Beijing, China
[4]Beijing National Research Center for Information Science and Technology, Beijing, China

## Abstract

Large Language Models (LLMs) are expected to possess comprehensive medical knowledge to support real-world clinical applications. While domain-specific fine-tuning effectively injects medical knowledge into LLMs, it often causes catastrophic forgetting of previously acquired knowledge and instruction-following capabilities. In this paper, we investigate this issue and reveal a pattern of proximity-dependent forgetting: knowledge that is semantically or topically close to the injected content is more likely to be forgotten, while unrelated knowledge shows minimal degradation. Moreover, we observe that existing mitigation techniques fail to address this type of forgetting effectively. Motivated by this observation and inspired by human learning mechanisms, we propose **InternAL** (**Inter**nal **Kn**owledge **A**ugmentation **L**earning), a novel approach that leverages LLMs' own internal knowledge to mitigate forgetting. InternAL first probes internal knowledge closely related to the injection by prompting the model with questions derived from the injected knowledge. This knowledge is then used to augment the original injection dataset, guiding the model to retain related prior knowledge during training. Experimental results on multiple LLMs (LLaMA, Qwen) demonstrate that InternAL significantly mitigates proximity-related forgetting while maintaining strong knowledge injection performance. Our findings provide new insights into the nature of catastrophic forgetting in medical knowledge injection and highlight a promising direction for robust domain adaptation in LLMs. Code and datasets are available at `https://github.com/THUMLP/InternAL`.

## 1   Introduction

Large language models (LLMs) achieve remarkable success across a wide range of domains [1–5] and exhibit great potential in specialized fields such as medicine. However, unlike general tasks, solving real-world clinical problems demands a deep understanding of domain-specific knowledge. While general-domain LLMs encode substantial world knowledge through pretraining and perform well on certain medical benchmarks [6, 7], recent studies [8, 9] suggest that their medical knowledge remains inadequate for supporting real-world clinical applications. Such findings highlight the need for effective strategies to inject essential medical knowledge into LLMs.

Existing post-pretraining knowledge injection methods can be broadly categorized into *Inference-time injection* and *Fine-tuning-based injection*. Inference-time injection methods [10–13], often realized through Retrieval-Augmented Generation (RAG), retrieve relevant knowledge from external sources and integrate it into the model's inference process. These methods effectively provide up-to-date

---

[*]Corresponding author.

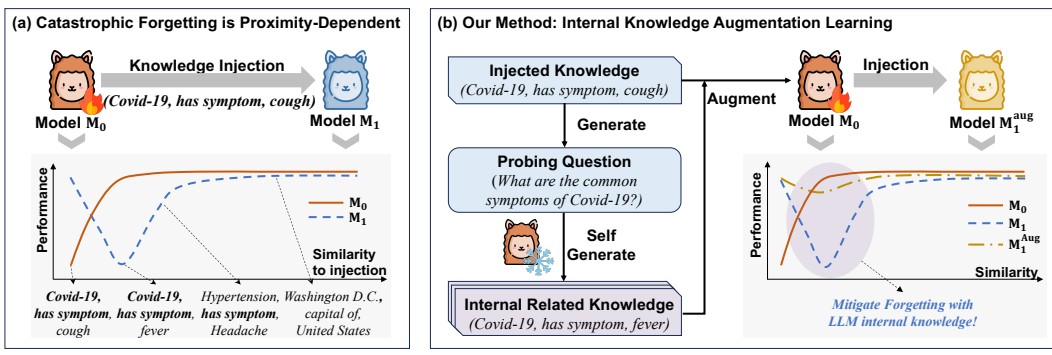

Figure 1: Left: Catastrophic forgetting exhibits a proximity-dependent pattern: knowledge closely related to the injected knowledge is more likely to be forgotten; Right: Our proposed **Inter**nal **Kn**owledge **A**ugmentation **L**earning (**InternAL**) method for mitigating catastrophic forgetting.

knowledge to LLMs, but their performance heavily relies on the quality of the retrieved content and may fail when the knowledge is required implicitly. On the other hand, fine-tuning-based injection methods [7, 14–16] train the model on datasets containing the target knowledge through fine-tuning, enabling the model to effectively apply the injected knowledge both explicitly and implicitly. However, such methods often suffer from *catastrophic forgetting* [17], where the model forgets previously acquired knowledge and instruction-following abilities after fine-tuning. Though several methods [18–24] have been proposed to mitigate catastrophic forgetting, their effectiveness in the medical domain has not been thoroughly investigated.

In this paper, we focus on fine-tuning-based methods and investigate the problem of catastrophic forgetting in medical knowledge injection. Specifically, we aim to address the following research questions: **RQ1**: *What kind of knowledge is more likely to be forgotten during knowledge injection?* **RQ2**: *How effective are existing methods in mitigating catastrophic forgetting?* and **RQ3**: *How to effectively mitigate catastrophic forgetting in medical knowledge injection?*

To answer these three research questions, we first conduct medical knowledge injection based on approximately 20k triples covering 21 types of medical knowledge extracted from the large-scale medical knowledge graph PrimeKG [25], and evaluate the injected model on a series of general and medical benchmarks. We observe that catastrophic forgetting exhibits a *proximity-dependent* pattern (illustrated in Figure 1a): **knowledge closely related to the injected knowledge is more prone to forgetting, while knowledge that is more distant tends to be less affected.** Moreover, existing mitigation methods show limited effectiveness, especially in preserving knowledge that is highly related to the injected content.

Motivated by these findings, we further propose a novel method (depicted in Figure 1b) called **Inter**nal **Kn**owledge **A**ugmentation **L**earning (**InternAL**), as a first attempt to mitigate catastrophic forgetting by leveraging the related internal knowledge of the target LLM. Specifically, we first extract relevant knowledge from the target LLM by prompting the LLM with questions generated based on the injected knowledge. We then incorporate this retrieved internal knowledge into the original injection dataset and fine-tune the model on the augmented data, thereby improving its ability to retain prior knowledge that is closely related to the injected content. The experimental results on several representative LLMs (e.g., LLaMA, Qwen) demonstrate that our method significantly mitigates forgetting of prior knowledge, particularly for knowledge that is closely associated with the injected content. Our contributions can be summarized as follows:

- We investigate the problem of catastrophic forgetting in medical knowledge injection and reveal a **proximity-dependent** forgetting pattern, where knowledge closely related to the injected knowledge is more likely to be forgotten.

- We evaluate several existing methods for mitigating catastrophic forgetting and find that they are not effective enough in the medical domain, especially in retaining relevant medical knowledge.

- We propose **InternAL**, a novel method that augments the injection process with internally retrieved knowledge from the LLM itself. Our method significantly alleviates forgetting, especially for knowledge that is semantically proximate to the injected content.

## 2 Related Work

**Knowledge Injection**   Existing knowledge injection methods can be categorized into the following two types: (1) *Inference-time injection* [10–13] (i.e. RAG) methods incorporate knowledge retrieved from external sources at inference time, enabling LLMs to access up-to-date information without additional fine-tuning. However, applying RAG in the medical domain presents several challenges, such as the difficulty of aligning queries with domain-specific content and the inability to retrieve or represent implicit knowledge that is required in many clinical reasoning tasks; (2) *Fine-tuning-based injection*[7, 14–16] methods train LLMs on datasets containing the target knowledge through fine-tuning. However, these methods often lead to catastrophic forgetting on prior knowledge and instruction-following abilities. In this paper, we aim to investigate and mitigate the catastrophic forgetting problem in the medical domain.

**Mitigating Catastrophic Forgetting**   Existing studies on mitigating catastrophic forgetting can be categorized into three types: (1) Replay-based methods[18, 19], which alleviate catastrophic forgetting by replaying old knowledge during training. This is typically achieved by mixing knowledge-injection samples with original training data; (2) Parameter-Efficient Fine-Tuning (PEFT) methods[20, 21], which mitigate forgetting by freezing most of the model parameters and updating only a small subset during fine-tuning; (3) Knowledge editing methods [22–24], which aim to inject new knowledge by first locating the relevant representation regions in the model and then performing small-scale parameter updates in those regions. In this work, we further investigate the effectiveness of these methods in mitigating catastrophic forgetting within the medical domain.

**Internal Knowledge Awakening**   There are also some studies that activate the internal knowledge of LLMs to improve their performance on knowledge-intensive tasks. These methods typically leverage prompting techniques [26] or fine-tuned small language models [27] to guide the LLMs to recall and utilize their internal knowledge in the reasoning process. The main difference between these methods and our work is that these methods focus on improving the model's performance on knowledge-intensive tasks, while our work aims to mitigate catastrophic forgetting by augmenting the knowledge injection process with relevant internal knowledge.

## 3 Catastrophic Forgetting is Proximity-Dependent

To mitigate catastrophic forgetting during medical knowledge injection, it is essential to first investigate which types of knowledge are more susceptible to forgetting (**RQ1**) and whether existing mitigation strategies are effective enough in this domain (**RQ2**). We begin by formulating the problem, and then describe our experimental setup, results and detailed analysis to answer these questions.

### 3.1 Problem Formulation

Suppose we are given an LLM $\mathcal{M}_0$ and a set of knowledge triplets $\mathcal{K}_{\text{inject}} = \{(h_i, r_i, t_i)\}_{i=1}^N$ to be injected, where $h_i$, $r_i$, and $t_i$ denote the head entity, relation, and tail entity of the $i^{\text{th}}$ triple, respectively. The basic optimization objective of a fine-tuning-based knowledge injection process can then be formulated as follows:

$$\mathcal{M}_1 = f_{\text{inject}}(\mathcal{M}_0; \mathcal{K}_{\text{inject}}) = \arg\max_{\mathcal{M}} \frac{1}{N} \sum_{i=1}^N \log\left(P_{\mathcal{M}}\left(t_i | h_i, r_i\right)\right) \tag{1}$$

where $f_{\text{inject}}$ is the knowledge injection process, $P_{\mathcal{M}}\left(t_i | h_i, r_i\right)$ is the probability of predicting the tail entity $t_i$ given the head entity $h_i$ and relation $r_i$ using the model $\mathcal{M}$. We denote the model after injection as $\mathcal{M}_1$. To measure the forgetting of prior knowledge caused by the knowledge injection process, we can evaluate the model $\mathcal{M}_1$ on an external benchmark $\mathcal{D}_{\text{test}}$, which contains $M$ test samples $\{(x_j, y_j)\}_{j=1}^M$. Then the catastrophic forgetting of prior knowledge can be measured by:

$$\mathrm{F}(\mathcal{M}_1 | \mathcal{M}_0; \mathcal{D}_{\text{test}}) = \mathrm{S}_{\mathcal{D}_{\text{test}}}(\mathcal{M}_1) - \mathrm{S}_{\mathcal{D}_{\text{test}}}(\mathcal{M}_0) \tag{2}$$

$$\mathrm{RF}(\mathcal{M}_1 | \mathcal{M}_0; \mathcal{D}_{\text{test}}) = \frac{\mathrm{F}(\mathcal{M}_1 | \mathcal{M}_0)}{\mathrm{S}_{\mathcal{D}_{\text{test}}}(\mathcal{M}_0)} = \frac{\mathrm{S}_{\mathcal{D}_{\text{test}}}(\mathcal{M}_0) - \mathrm{S}_{\mathcal{D}_{\text{test}}}(\mathcal{M}_1)}{\mathrm{S}_{\mathcal{D}_{\text{test}}}(\mathcal{M}_0)} \tag{3}$$

where $S_{\mathcal{D}_{\text{test}}}(\mathcal{M})$ is the performance (e.g., accuracy, f1-score, etc.) of the model $\mathcal{M}$ on the test dataset $\mathcal{D}_{\text{test}}$, $F(\mathcal{M}_1|\mathcal{M}_0)$ denotes the absolute forgetting of prior knowledge, and $RF(\mathcal{M}_1|\mathcal{M}_0)$ denotes the relative forgetting–i.e., the proportion of performance drop relative to the original model.

One of our core questions (RQ1) is to investigate what types of knowledge are more vulnerable to forgetting during medical knowledge injection. Prior work [22, 23] has shown that knowledge representations in LLMs exhibit locality, where highly-related facts tend to share representation space. Inspired by this, we hypothesize that the proximity between the injected knowledge and the knowledge embedded in the test set $\mathcal{D}_{\text{test}}$, denoted by $\text{Sim}(\mathcal{K}_{\text{inject}}, \mathcal{K}_{\text{test}})$, significantly influences the extent of forgetting $F(\mathcal{M}_1|\mathcal{M}_0; \mathcal{D}_{\text{test}})$ on the test set. We will validate this hypothesis through experiments in the following sections.

## 3.2 Experimental Setup

**Datasets For Knowledge Injection**  We leverage PrimeKG [25], a comprehensive biomedical knowledge graph that integrates knowledge from 20 curated biomedical knowledge bases (e.g., UMLS [28], DrugBank [29]). PrimeKG encompasses over 4 million triples spanning 29 diverse types of medical knowledge, making it a rich and representative resource for medical knowledge injection into LLMs. In our study, we select 21 important categories of medical knowledge, such as disease phenotypes, drug indications/contraindications/side effects, and protein functions/interactions. Considering the large scale of PrimeKG, we randomly sampled $\sim 1k$ triples for each type of knowledge, resulting in a total of 20,864 triples. To identify which knowledge should be injected and to evaluate the effectiveness of the injection, we generate $k_{\text{test}}$ four-option multiple-choice questions ($\{q_i^j\}_{j=1}^{k_{\text{test}}}$) for each sampled knowledge triple $z_i$[2]. These questions are designed to evaluate the model's basic understanding of the corresponding knowledge. We then evaluate the original model $\mathcal{M}_0$ using these questions and measure its accuracy $\text{Acc}_i(\mathcal{M}_0)$ on each knowledge triple $z_i$:

$$\text{Acc}_i(\mathcal{M}) = \frac{1}{k_{\text{test}}} \sum_{j=1}^{k_{\text{test}}} I(p_j^i(\mathcal{M}) = l_j^i) \tag{4}$$

where $p_j^i(\mathcal{M})$ is the predicted answer of the model $\mathcal{M}$ for the $j^{\text{th}}$ question $q_i^j$ corresponding to $z_i$, $l_j^i$ is the label of $q_i^j$, and $I(\cdot)$ is the indicator function. Knowledge triples with an accuracy below 0.25 (i.e., lower than random guessing on 4-option questions) are selected to construct the injection set $\mathcal{K}_{\text{inject}}$. Based on this, we construct a corresponding test set $\mathcal{D}_{\text{inject}} = \{q_i^j | z_i \in \mathcal{K}_{\text{inject}}, 1 \leq j \leq k_{\text{test}}\}$ to evaluate the effectiveness of knowledge injection. We also create two complementary test sets: (1) triples with accuracies above 0.75 are used to build a test set $\mathcal{D}_{\text{eval}}$ for assessing knowledge forgetting; and (2) based on all 20,864 sampled knowledge triples, we further construct a comprehensive test set $\mathcal{D}_{\text{total}}$ to evaluate the overall effectiveness of the knowledge injection process. Further details on the construction process and statistics of the injected dataset are provided in appendix A.

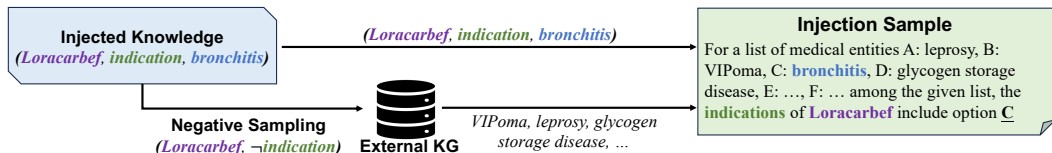

Figure 2: An overview of the Reference-style Knowledge Injection (**RefInject**) method.

**Knowledge Injection Method**  We develop a knowledge injection method named Referencing-style Knowledge Injection (**RefInject**) that converts structured knowledge triples into natural language instances suitable for fine-tuning. Specifically, for each triple $z_i = (h_i, r_i, t_i)$, we sample $m - 1$ negative tail entities $t_1^{\text{neg}}, t_2^{\text{neg}}, \cdots, t_{m-1}^{\text{neg}}$ from PrimeKG, and construct a referencing-style demonstration (see Figure 2). The LLM learns to predict the correct option (underlined in the figure) corresponding to the ground-truth tail entity $t_i$ among $m$ candidates. To prevent the model from exploiting superficial patterns (e.g., entity co-occurrence), we follow the method proposed in [30]

---

[2]We generate multiple test questions for each knowledge triplet to ensure the robustness of evaluation results.

and generate $k$ diverse samples for each triple, with the correct answer randomly assigned to different positions across samples. We set $m = 10$ and $k = 20$ in our experiments, as larger values yield diminishing returns. More details (e.g., hyperparameters) can be found in appendix B.

**Baselines For Mitigating Catastrophic Forgetting**  We construct representative baseline methods[3] to mitigate catastrophic forgetting during knowledge injection, including: (1) General-domain Fine-Tuning (GenFT): continual fine-tuning on general-domain instruction data to restore instruction-following ability. In our study, we use the MMLU development set (285 examples) for SFT; (2) Parameter-Efficient Fine-Tuning (PEFT): we apply LoRA[21], which updates only a small subset of parameters; (3) Knowledge Editing: we adopt MEMIT [23] and AlphaEdit [24], both achieving state-of-the-art performance in editing factual knowledge. More implementation details of the baseline methods (e.g., hyperparameters setting, training epochs, etc.) are provided in appendix C.

**Evaluation Benchmarks**  To investigate what type of knowledge is more likely to be forgotten, we evaluate the forgetting of the injected model on a series of general and medical benchmarks. Specifically, we leverage MMLU [31] (Non-medical subset, denoted as MMLU-O), ARC-challenge [32] (ARC-C) and CommonSenseQA [33] (CSQA) to evaluate the model's performance on general knowledge. For the medical domain, we utilize MedQA [34], MMLU medical subset (MMLU-Med). The details of these benchmarks as well as the evaluation settings (prompt formats, inference hyperparameters) are provided in appendices D and E.

**Backbone Models**  We primarily conduct experiments on four well-known LLMs: Llama3-8B [3] and Qwen3 1.7B, 8B, and 32B [35], chosen for their availability and strong performance on a range of general-domain tasks. Due to resource constraints, we conduct full-parameter fine-tuning only on the smaller models (Llama3-8B and Qwen3-1.7, 8B), and apply LoRA to the largest model (Qwen3-32B). In our study, we use the instruction-tuned version of these models.

## 3.3  Results and Analysis

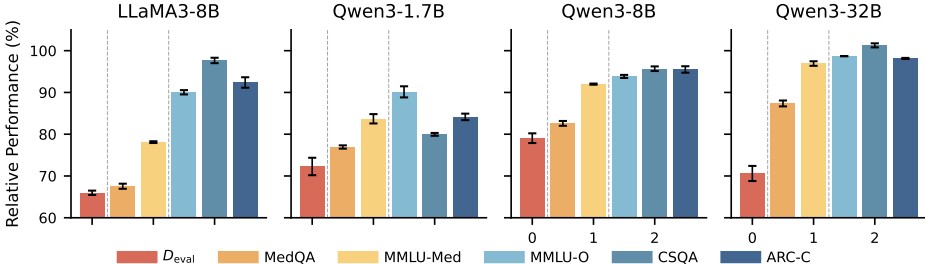

Figure 3: Relative performance (%) of LLMs on evaluation benchmarks after knowledge injection, normalized to their original performance. Error bars represent the standard deviation across 3 runs.

**Knowledge Closer to Injected Facts Is More Prone to Forgetting**  We perform knowledge injection on the selected LLMs using the RefInject method and evaluate forgetting across evaluation benchmarks. The relative performance (normalized to the performance before injection) of the injected models is shown in Figure 3. We observe that the performance of the injected models on all the medical benchmarks ($\mathcal{D}_{\text{eval}}$, MedQA and MMLU-Med) drops significantly, while the performance on general benchmarks (MMLU-O, ARC-C, CSQA) remains relatively stable. For example, Llama3-8B experiences a >30% drop in relative performance on the MedQA benchmark, while it retains over 90% of its original performance on all the general benchmarks. Such phenomenon indicates that LLMs are prone to forgetting knowledge in domains closely related to the injected content—such as the medical domain in this study—during the injection process.

To investigate the phenomenon of proximity-dependent forgetting in a more fine-grained view, we divide the medical benchmarks into proximal and relatively distal sets based on the knowledge

---

[3]Replay-based methods are not applicable in our study, because the old data for pretraining and instruction fine-tuning existing LLMs is typically not publicly available.

proximity between the injected knowledge set $\mathcal{D}_{\mathrm{inject}}$ and the evaluation samples. For the PrimeKG-based evaluation set ($\mathcal{D}_{\mathrm{eval}}$), samples are classified as proximal if they share the same head entity and relation with any injected knowledge triple; otherwise, they are considered distal. For MedQA and MMLU-Med, where explicit knowledge triples per test sample are unavailable, we embed both the test samples and the injected knowledge into a shared space using the MedEmbed model [36], and compute cosine similarity to estimate proximity. A threshold of 0.8 is then used to select samples that are considered proximal. The detailed splitting process is provided in appendix F.

Table 1: Performance (%) of the original and injected models on medical benchmarks, divided according to the proximity to the injected knowledge.

| Model | $\mathcal{D}_{\mathrm{eval}}$ | | MedQA | | MMLU-Med | |
|---|---|---|---|---|---|---|
| | Proximal | Distal | Proximal | Distal | Proximal | Distal |
| Llama3-8B | 88.9 | 91.9 | 56.0 | 48.8 | 84.0 | 68.1 |
| +Knowledge Injection | $51.2_{\downarrow \mathbf{42.4}\%}$ | $61.9_{\downarrow 32.6\%}$ | $35.1_{\downarrow \mathbf{37.4}\%}$ | $34.0_{\downarrow 30.4\%}$ | $64.1_{\downarrow \mathbf{23.7}\%}$ | $53.4_{\downarrow 21.6\%}$ |
| Qwen3-1.7B | 86.8 | 89.2 | 40.3 | 36.4 | 68.9 | 57.7 |
| +Knowledge Injection | $56.2_{\downarrow \mathbf{35.3}\%}$ | $66.0_{\downarrow 26.0\%}$ | $29.0_{\downarrow \mathbf{28.0}\%}$ | $27.9_{\downarrow 23.4\%}$ | $49.9_{\downarrow \mathbf{27.6}\%}$ | $48.5_{\downarrow 15.8\%}$ |
| Qwen3-8B | 88.7 | 91.8 | 64.7 | 56.1 | 92.2 | 77.4 |
| +Knowledge Injection | $64.0_{\downarrow \mathbf{27.8}\%}$ | $74.1_{\downarrow 19.3\%}$ | $51.6_{\downarrow \mathbf{20.3}\%}$ | $47.1_{\downarrow 16.2\%}$ | $84.1_{\downarrow \mathbf{8.8}\%}$ | $71.4_{\downarrow 7.8\%}$ |
| Qwen3-32B | 89.0 | 92.8 | 72.2 | 67.7 | 93.0 | 80.4 |
| +Knowledge Injection | $63.3_{\downarrow 28.8\%}$ | $65.5_{\downarrow \mathbf{29.5}\%}$ | $59.6_{\downarrow \mathbf{17.5}\%}$ | $59.3_{\downarrow 12.4\%}$ | $80.0_{\downarrow \mathbf{14.0}\%}$ | $74.7_{\downarrow 7.0\%}$ |

Table 1 presents the performance of the original and injected models on the medical benchmarks, divided into proximal and distal subsets, with subscripts showing the relative forgetting. Across all datasets, we observe that the forgetting of proximal knowledge is generally more severe than that of distal knowledge. For example, in the case of Llama3-8B, the relative forgetting on proximal knowledge is 42.4, 37.4, and 23.7 across benchmarks, while it is only 32.6, 30.4 and 21.6 for distal knowledge. These results validate our hypothesis that **the proximity between the injected knowledge and the test samples significantly influences the extent of forgetting**, and that knowledge that is more closely related to the injected knowledge is more likely to be forgotten.

Table 2: Performance (%) of the original (Llama3-8B) and injected models using various methods on the medical and general benchmarks.

| Model | Medical | | | | | General | | |
|---|---|---|---|---|---|---|---|---|
| | $\mathcal{D}_{\mathrm{total}}$ | $\mathcal{D}_{\mathrm{inject}}$ | $\mathcal{D}_{\mathrm{eval}}$ | MedQA | MMLU-Med | MMLU-O | ARC-C | CSQA |
| Original | 51.5 | 9.7 | 91.4 | 50.7 | 69.8 | 59.8 | 75.4 | 66.4 |
| MEMIT | 53.4 | 36.9 | 75.9 | 48.0 | 66.2 | 58.3 | 75.0 | 65.3 |
| AlphaEdit | 52.3 | 32.7 | 77.1 | 44.7 | 64.9 | 57.4 | 73.9 | 64.8 |
| RefInject | 65.0 | 77.4 | 60.3 | 34.2 | 54.5 | 53.8 | 69.7 | 64.9 |
| +LoRA | 66.9 | 75.9 | 65.3 | 36.7 | 55.3 | 55.6 | 72.1 | 65.0 |
| +GenFT | 68.8 | 73.4 | 71.4 | 41.8 | 64.0 | 59.6 | 76.0 | 69.3 |

**Existing Mitigation Methods Are Not Effective Enough for Knowledge Closely Related to Injected Knowledge** We further investigate the effectiveness of methods for mitigating catastrophic forgetting in the knowledge injection process, with the results on Llama3-8B summarized in Table 2 (full results are provided in appendix G). While knowledge editing methods (MEMIT, AlphaEdit) retain original knowledge well, their performance on injected knowledge is poor (36.9 and 32.7 on $\mathcal{D}_{\mathrm{inject}}$), resulting in limited overall injection effectiveness (+1.9 and +0.8 on $\mathcal{D}_{\mathrm{total}}$). A possible reason is that these methods modify only a limited number of model parameters, which may insufficient for enabling LLMs to generalize the injected knowledge effectively. In contrast, LoRA and GenFT retain most of RefInject's injection effectiveness, achieving accuracies of 75.9 and 73.4 on $\mathcal{D}_{\mathrm{inject}}$ and overall performance of 66.9 and 68.8 on $\mathcal{D}_{\mathrm{total}}$, respectively. While these approaches also mitigate forgetting of the original knowledge to a certain degree, notable degradation remains, particularly on medical benchmarks. Notably, fine-tuning with general-domain instruction data (GenFT) effectively

restores most of performance on general-domain datasets, but forgetting on medical benchmarks persists (e.g., 41.8 vs. 50.7 on MedQA). Our findings suggest that though the catastrophic forgetting of knowledge injection can be mitigated to some extent by existing methods, they are not effective enough regarding either the injection effectiveness or the retention of original knowledge.

# 4 Mitigating Catastrophic Forgetting via LLMs' Internal Knowledge

## 4.1 Methodology

**Schema of Internal Knowledge Augmentation** In this section, we propose a novel method called Internal Knowledge Augmentation Learning (**InternAL**) as a first attempt to mitigate catastrophic forgetting by leveraging related internal knowledge from the target LLM. An overview of the proposed method is presented in Figure 4. Our findings in the previous section indicate that knowledge more closely related to the injected content is particularly susceptible to forgetting. To address this, we first extract the relevant knowledge from the target model $\mathcal{M}_0$:

$$\mathcal{K}_{\text{inner}} = f_{\text{probe}}(\mathcal{M}_0; \mathcal{K}_{\text{inject}}) \tag{5}$$

where $f_{\text{probe}}$ is the probing function that extracts the internal knowledge relevant to $\mathcal{K}_{\text{inject}}$ from the model $\mathcal{M}_0$. Then, the original knowledge injection process can be augmented with the internal knowledge $\mathcal{K}_{\text{inner}}$:

$$\mathcal{M}_1^{aug} = f_{\text{inject}}^{aug}(\mathcal{M}_0; \mathcal{K}_{\text{inject}}, \mathcal{K}_{\text{inner}}) \tag{6}$$

By attending to the relevant internal knowledge during the injection process, the proposed InternAL method aims to mitigate the forgetting of the most relevant knowledge to the injected knowledge.

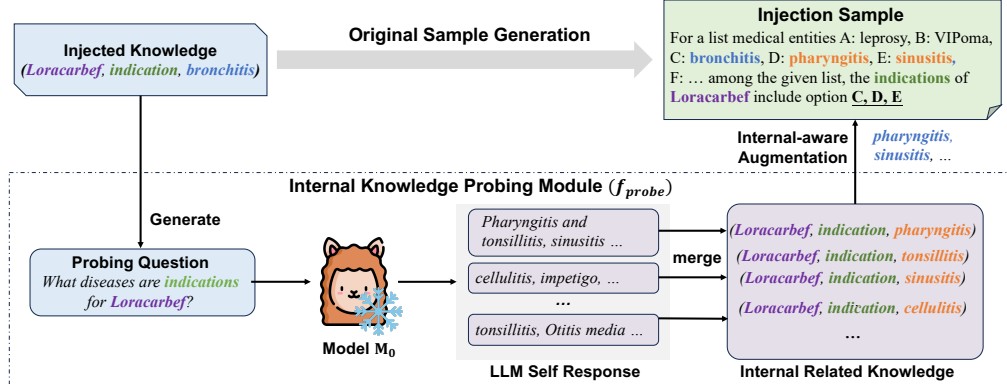

Figure 4: Overview of the proposed Internal Knowledge Augmentation Learning (**InternAL**) method.

**Internal Knowledge Probing** ($f_{\text{probe}}$) To extract the internal knowledge from the target LLM, we develop a internal knowledge probing module that first generates a probing question $Q_i$ based on the head entity $h_i$ and relation $r_i$ of the injected knowledge triple $(h_i, r_i, t_i)$ with templates. Such question is designed to prompt the model to recall all the possible tail entities that have the relation $r_i$ with the head entity $h_i$. Then we prompt the target LLM $\mathcal{M}_0$ using the probing question $K = 5$ times to generate $K$ different LLM responses $R_i^1, R_i^2, \cdots, R_i^K$. Finally, we extract the tail entities from the generated responses and filter out the duplicates to form the internal knowledge set $\mathcal{K}_{\text{inner}}$:

$$\mathcal{K}_{\text{inner}} = \{(h_i, r_i, t') | t' \in \bigcup_{k=1}^{K} f_{\text{extract}}(R_i^k), 1 \leq i \leq N\} \tag{7}$$

where $f_{\text{extract}}$ is the function that extracts the tail entities from the generated responses, implemented by prompting the target LLM. More details on the probing process, including the templates and generation hyperparameters, can be found in appendix H.

**Internal-aware Sample Augmentation($f_{\text{inject}}^{\text{aug}}$)** After extracted internal knowledge relevant to the injected knowledge, we augment the sample generation process with the extracted internal knowledge. Specifically, for each injected knowledge triple $z_i = (h_i, r_i, t_i)$, we sample $n$ relevant tail entities $t_1', t_2', \cdots, t_n'$ from $\mathcal{K}_{\text{inner}}$ that share the same head entity and relation with $z_i$, and sample $m - n - 1$ negative tail entities $t_1^{\text{neg}}, t_2^{\text{neg}}, \cdots, t_{m-n-1}^{\text{neg}}$ from PrimeKG. Similar to the RefInject method, we construct a referencing-style demonstration using the sampled tail entities. The LLM is trained to select multiple correct options from $m$ candidates, including both the injected tail entity $t_i$ and the relevant tail entities $t_1', t_2', \cdots, t_n'$. We keep the $m$ and $k$ consistent with RefInject, and random choose $n$ from 0 to 3 for each sample to prevent the model from learning statistical biases. The generated samples are then used to fine-tune the target LLM $\mathcal{M}_0$ to obtain the injected model $\mathcal{M}_1^{aug}$. Further details on the sample augmentation process can also be found in appendix H.

Table 3: Performance (%) of the baseline knowledge injection method (RefInject) and the proposed method (InternAL). The lowest relative forgetting on each benchmark is underlined.

| Model | Medical | | | | | General | | |
|---|---|---|---|---|---|---|---|---|
| | $\mathcal{D}_{\text{total}}$ | $\mathcal{D}_{\text{inject}}$ | $\mathcal{D}_{\text{eval}}$ | MedQA | MMLU-Med | MMLU-O | ARC-C | CSQA |
| Llama3-8B | 51.5 | 9.7 | 91.4 | 50.7 | 69.8 | 59.8 | 75.4 | 66.4 |
| +RefInject | 65.0 | 77.4 | 60.3$_{\downarrow 34.0\%}$ | 34.2$_{\downarrow 32.6\%}$ | 54.5$_{\downarrow 21.9\%}$ | 53.8$_{\downarrow 10.0\%}$ | 69.7$_{\downarrow 7.6\%}$ | 64.9$_{\downarrow 2.3\%}$ |
| +RefInject+GenFT | 68.8 | 73.4 | 71.4$_{\downarrow 21.9\%}$ | 41.8$_{\downarrow 17.6\%}$ | 64.0$_{\downarrow 8.3\%}$ | 59.6$_{\downarrow 0.2\%}$ | 76.0$_{\uparrow 0.7\%}$ | 69.3$_{\uparrow 4.4\%}$ |
| +InternAL (**ours**) | 69.3 | 74.3 | 70.9$_{\downarrow 22.4\%}$ | 39.5$_{\downarrow 22.2\%}$ | 56.8$_{\downarrow 18.7\%}$ | 54.7$_{\downarrow 8.5\%}$ | 70.3$_{\downarrow 6.9\%}$ | 60.9$_{\downarrow 8.3\%}$ |
| +InternAL+GenFT | **71.2** | 71.4 | 77.4$_{\downarrow 15.4\%}$ | 45.1$_{\downarrow 11.1\%}$ | 66.1$_{\downarrow 5.3\%}$ | 60.4$_{\uparrow 1.0\%}$ | 75.7$_{\downarrow 0.3\%}$ | 69.8$_{\uparrow 5.1\%}$ |
| Qwen3-1.7B | 42.6 | 9.7 | 88.7 | 37.5 | 59.0 | 52.5 | 71.6 | 66.4 |
| +RefInject | 60.4 | 63.0 | 64.1$_{\downarrow 27.7\%}$ | 28.8$_{\downarrow 23.1\%}$ | 49.4$_{\downarrow 16.3\%}$ | 47.3$_{\downarrow 9.9\%}$ | 60.2$_{\downarrow 15.9\%}$ | 53.1$_{\downarrow 20.0\%}$ |
| +RefInject+GenFT | 62.9 | 60.4 | 72.8$_{\downarrow 17.9\%}$ | 31.3$_{\downarrow 16.6\%}$ | 55.0$_{\downarrow 6.8\%}$ | 52.1$_{\downarrow 0.9\%}$ | 68.1$_{\downarrow 4.9\%}$ | 63.3$_{\downarrow 4.7\%}$ |
| +InternAL (**ours**) | 63.5 | 59.3 | 75.1$_{\downarrow 15.3\%}$ | 32.0$_{\downarrow 14.7\%}$ | 51.4$_{\downarrow 12.8\%}$ | 47.6$_{\downarrow 9.4\%}$ | 61.6$_{\downarrow 13.9\%}$ | 58.3$_{\downarrow 12.2\%}$ |
| +InternAL+GenFT | **65.1** | 58.7 | 79.2$_{\downarrow 10.8\%}$ | 33.3$_{\downarrow 11.1\%}$ | 58.0$_{\downarrow 1.7\%}$ | 52.0$_{\downarrow 1.1\%}$ | 68.5$_{\underline{\downarrow 4.3\%}}$ | 61.8$_{\downarrow 6.9\%}$ |
| Qwen3-8B | 49.3 | 9.4 | 91.4 | 58.5 | 79.0 | 67.2 | 87.3 | 80.3 |
| +RefInject | 68.6 | 72.1 | 72.2$_{\downarrow 21.0\%}$ | 48.3$_{\downarrow 17.4\%}$ | 72.7$_{\downarrow 8.0\%}$ | 63.0$_{\downarrow 6.2\%}$ | 83.4$_{\downarrow 4.5\%}$ | 76.8$_{\downarrow 4.3\%}$ |
| +RefInject+GenFT | 70.0 | 70.4 | 76.6$_{\downarrow 16.2\%}$ | 50.8$_{\downarrow 13.2\%}$ | 75.6$_{\downarrow 4.3\%}$ | 69.1$_{\uparrow 2.8\%}$ | 87.3$_{\downarrow 0.0\%}$ | 78.6$_{\downarrow 2.1\%}$ |
| +InternAL (**ours**) | 73.2 | 72.0 | 82.0$_{\downarrow 10.2\%}$ | 51.1$_{\downarrow 12.7\%}$ | 74.9$_{\downarrow 5.1\%}$ | 65.9$_{\downarrow 1.9\%}$ | 84.9$_{\downarrow 2.8\%}$ | 76.7$_{\downarrow 4.5\%}$ |
| +InternAL+GenFT | **73.7** | 70.8 | 84.0$_{\underline{\downarrow 8.0\%}}$ | 52.3$_{\downarrow 10.6\%}$ | 76.9$_{\downarrow 2.6\%}$ | 70.2$_{\uparrow 4.5\%}$ | 87.7$_{\uparrow 0.5\%}$ | 78.2$_{\downarrow 2.5\%}$ |
| Qwen3-32B | 59.3 | 10.2 | 92.4 | 68.2 | 80.8 | 68.5 | 86.9 | 83.5 |
| +RefInject | 63.4 | 69.4 | 65.2$_{\downarrow 29.4\%}$ | 59.6$_{\downarrow 12.6\%}$ | 78.3$_{\downarrow 3.1\%}$ | 67.6$_{\downarrow 1.3\%}$ | 85.3$_{\downarrow 1.9\%}$ | 84.6$_{\uparrow 1.3\%}$ |
| +RefInject+GenFT | 66.8 | 73.6 | 68.6$_{\downarrow 25.7\%}$ | 60.2$_{\downarrow 11.8\%}$ | 79.9$_{\downarrow 1.1\%}$ | 70.9$_{\uparrow 3.5\%}$ | 88.8$_{\uparrow 2.2\%}$ | 84.1$_{\uparrow 0.7\%}$ |
| +InternAL (**ours**) | 66.5 | 64.7 | 72.8$_{\downarrow 21.1\%}$ | 63.3$_{\downarrow 7.1\%}$ | 79.3$_{\downarrow 1.9\%}$ | 67.7$_{\downarrow 1.3\%}$ | 86.9$_{\downarrow 0.1\%}$ | 82.8$_{\downarrow 0.9\%}$ |
| +InternAL+GenFT | **73.5** | 67.7 | 83.1$_{\underline{\downarrow 10.0\%}}$ | 63.4$_{\underline{\downarrow 7.0\%}}$ | 81.9$_{\uparrow 1.3\%}$ | 72.6$_{\uparrow 5.9\%}$ | 89.9$_{\underline{\uparrow 3.4\%}}$ | 83.0$_{\downarrow 0.7\%}$ |

## 4.2 Results and Analysis

**Effectiveness of InternAL across Benchmarks** We conduct experiments to evaluate the effectiveness of the proposed InternAL method in mitigating catastrophic forgetting during knowledge injection, comparing it against the original RefInject method, both with and without general-domain Fine-Tuning (GenFT). The results are presented in Table 3. We observe that InternAL significantly alleviates catastrophic forgetting across all medical benchmarks and backbone models, while maintaining stable performance on general-domain benchmarks. For instance, on the MedQA benchmark, Llama3-8B fine-tuned with InternAL reduces relative forgetting by 10.4 compared to the original RefInject method (22.2 vs. 32.6). Furthermore, InternAL can be combined with GenFT to further mitigate the forgetting of original knowledge. For example, Llama3-8B achieves 77.4 on $\mathcal{D}_{\text{eval}}$ after applying InternAL+GenFT, reducing relative forgetting by 6.5 compared to RefInject+GenFT (15.4 vs. 21.9). These results suggest that while general-domain instruction tuning (GenFT) effectively restores the model's instruction-following capability, applying the proposed InternAL method provides additional gains in preserving the original knowledge.

**Effectiveness on Proximal vs. Distal Knowledge** We investigate the effectiveness of the proposed InternAL method on the proximal and distal subsets of the medical benchmarks, with results presented in Table 4. We observe that though InternAL can effectively mitigate forgetting on both proximal

Table 4: Performance (%) of RefInject and InternAL on proximal and distal subsets of medical benchmarks (using Llama3-8B as backbone).

| Model | $\mathcal{D}_{\text{eval}}$ | | MedQA | | MMLU-Med | |
|---|---|---|---|---|---|---|
| | Proximal | Distal | Proximal | Distal | Proximal | Distal |
| Llama3-8B | 88.9 | 91.9 | 56.0 | 48.8 | 84.0 | 68.1 |
| +RefInject | $51.2_{\downarrow42.4\%}$ | $61.9_{\downarrow32.6\%}$ | $35.1_{\downarrow37.4\%}$ | $33.9_{\downarrow30.6\%}$ | $64.1_{\downarrow23.7\%}$ | $53.4_{\downarrow21.6\%}$ |
| +RefInject+GenFT | $62.4_{\downarrow29.8\%}$ | $72.9_{\downarrow20.6\%}$ | $45.8_{\downarrow18.2\%}$ | $40.3_{\downarrow17.4\%}$ | $76.7_{\downarrow8.7\%}$ | $62.5_{\downarrow8.3\%}$ |
| +InternAL (**ours**) | $63.6_{\downarrow28.5\%}$ | $72.2_{\downarrow21.4\%}$ | $43.1_{\downarrow23.2\%}$ | $38.2_{\downarrow21.9\%}$ | $66.7_{\downarrow20.7\%}$ | $55.6_{\downarrow18.4\%}$ |
| +InternAL+GenFT | $71.2_{\downarrow19.9\%}$ | $78.4_{\downarrow14.7\%}$ | $50.0_{\downarrow10.7\%}$ | $43.3_{\downarrow11.3\%}$ | $80.3_{\downarrow4.5\%}$ | $64.4_{\downarrow5.4\%}$ |

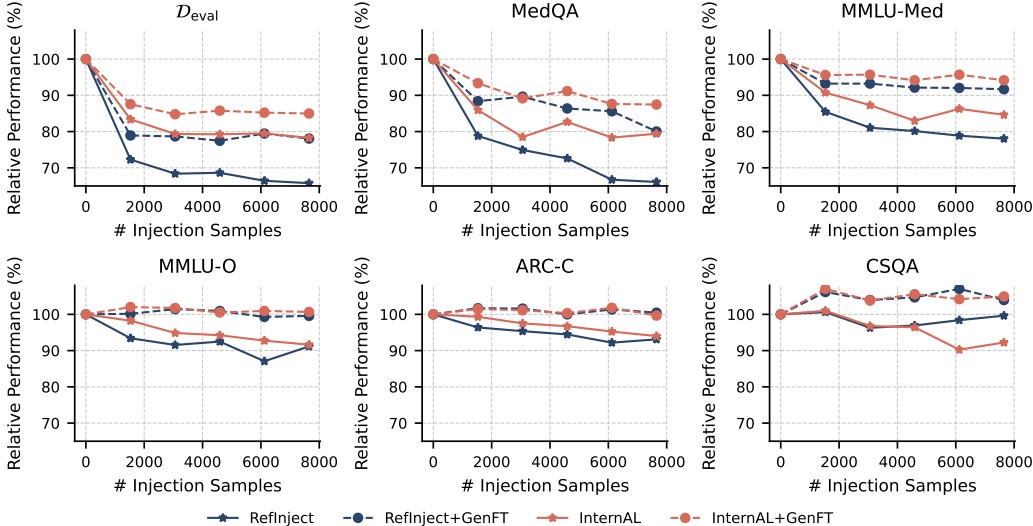

Figure 5: Relative performance (%) of Llama3-8B trained with different knowledge injection methods on various evaluation benchmarks, with varying numbers of injected knowledge triples. All results are normalized to the model's performance prior to injection.

and distal subsets, the mitigating effect is more pronounced on the proximal subset. For example, on the MedQA benchmark, InternAL reduces relative forgetting by 14.2 on the proximal subset (37.4 vs. 23.2) and 8.7 on the distal subset (30.6 vs. 21.9). This indicates that the proposed method is particularly effective in preserving the knowledge that is more closely related to the injected knowledge, which is consistent with our hypothesis.

**Effectiveness across Injection Scale** We further validate the proposed InternAL method across different scales of knowledge injection by conducting experiments with varying ratio of injected knowledge to the original injection set $\mathcal{K}_{\text{inject}}$ (i.e., 0.2, 0.4, 0.6, 0.8). The results presented in Figure 5 show that the proposed InternAL method generally outperforms the original RefInject method across all scales of knowledge injection and the performance drops much slower than RefInject on medical benchmarks. This indicates that internal knowledge augmentation better preserves essential medical knowledge as the scale of knowledge injection increases.

**Representation-Level Analysis** To further understand how InternAL mitigates catastrophic forgetting, we analyze the representation changes before and after knowledge injection based on Llama3-8B and Qwen3-8B. Then, we compute the average representation shift on the evaluation set $\mathcal{D}_{\text{eval}}$ to quantify the extent of representation change caused by knowledge injection on the unlearned knowledge. The results are presented in Figure 6. We observe that InternAL consistently results in a smaller average representation shift than RefInject, especially on the middle layers, which are known to capture more knowledge-related information [23]. This suggests that InternAL effectively preserves

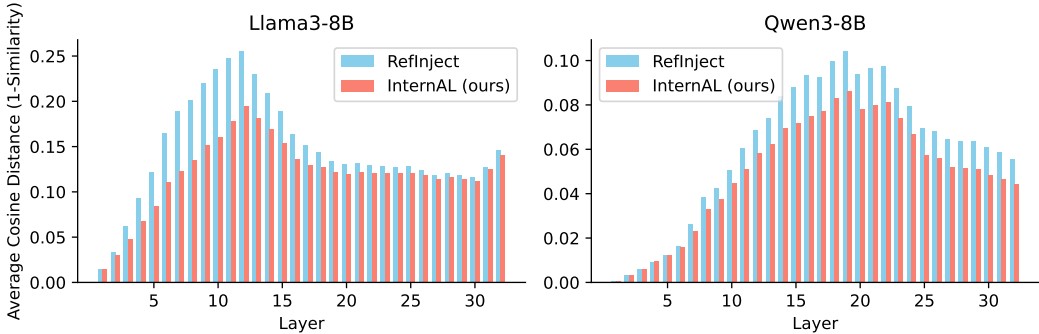

Figure 6: Average representation shift on the evaluation set $\mathcal{D}_{\mathrm{eval}}$ after knowledge injection using RefInject and InternAL methods.

the original knowledge representations during the injection process, thereby mitigating the forgetting.

### 4.3 Extended Discussion

**Impact on Hallucination Level** Although the augmented knowledge used in InternAL is generated by the target LLM itself and may contain factual errors, its hallucination level is inherently limited by the model and therefore does not introduce additional hallucinations. To validate this, we conduct a human evaluation on the hallucination level before and after knowledge injection, and the results in appendix I show that the proposed method does not further increase the hallucination level.

**Generalizability to Other Domain** While our study mainly focus on the medical domain, the proposed InternAL method may also be applicable to other domains. To verify this, we conducted a small-scale study in the human geography domain. Results in appendix J show that InternAL effectively mitigates catastrophic forgetting during knowledge injection in this domain as well, indicating its potential generalizability.

**Generalizability to Other Knowledge Formats** While our method primarily focuses on structured knowledge, it can also be extended to other formats, such as unstructured texts. To test this, we conducted a preliminary study on free-form medical texts (e.g., clinical guidelines) and adapted InternAL for this setting. Results in appendix K show that InternAL effectively mitigates catastrophic forgetting here as well, demonstrating its potential generalizability on unstructured data.

## 5 Conclusion

In this paper, we explore the challenge of catastrophic forgetting in large language models during medical knowledge injection. Our experiments reveal a clear proximity-dependent forgetting phenomenon: knowledge that is semantically or topically close to the injected content is more prone to be forgotten. We evaluate several existing mitigation techniques and find them inadequate in preserving knowledge that is closely related to the injected knowledge. To address this, we propose Internal Knowledge Augmentation Learning (InternAL), a novel approach that leverages the LLMs' internal knowledge to enhance the injection process. Experimental results show that InternAL consistently mitigates forgetting across diverse medical benchmarks while preserving most of the injection effectiveness. We hope our findings shed light on the underlying properties of catastrophic forgetting in medical knowledge injection and highlight a promising direction for future work that harnesses LLMs' internal knowledge to address this issue.

**Limitations.** There are two main limitations in our work. First, our study mainly focus on the catastrophic forgetting in the medical domain, and the behavior of catastrophic forgetting in other domains may differ (though we have conducted some preliminary studies in other domains as discussed in appendix J). Second, while we propose a novel method to mitigate catastrophic forgetting, it is still far from completely resolving the catastrophic forgetting problem. Future work should focus on extracting more relevant internal knowledge and developing more effective augmentation methods to mitigate catastrophic forgetting in knowledge injection.

## Acknowledgments

We thank the anonymous reviewers for their insightful comments and suggestions. This work was supported by Noncommunicable Chronic Diseases-National Science and Technology Major Project (Grant No. 2023ZD0506501).

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

# A    Details of Dataset for Knowledge Injection

PrimeKG[4] is a large-scale biomedical knowledge graph that contains over 4 million triples, covering a wide range of medical concepts and relationships. It is constructed from 20 different biomedical knowledge bases, including UMLS, DrugBank, OMIM, and others. In our study, we utilize PrimeKG to construct dataset for knowledge injection and evaluation of catastrophic forgetting. An overview of the dataset construction process is shown in Figure 7:

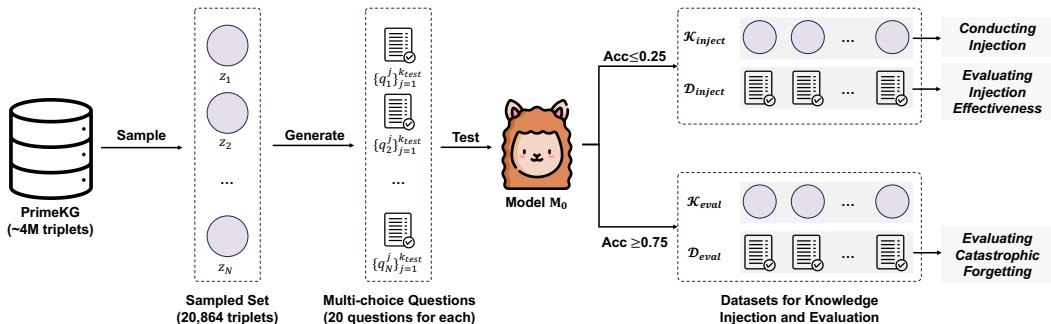

Figure 7: An overview of the dataset construction process based on PrimeKG.

To support our study, we selected 21 crucial knowledge types from PrimeKG as listed in Table 6, and randomly sampled 1,000 triples from each type given the large scale of PrimeKG. To identify knowledge not well acquired by the LLM prior to injection, we first generated multiple-choice questions (MCQs) for each sampled triplet and evaluated the original model $\mathcal{M}_0$ based on its performance. An example of the question generation process is shown in Figure 8, with templates provided in Table 5. For each triplet, we created $k_{\text{test}} = 20$ questions, each comprising one correct answer (the triplet's tail) and three distractors randomly sampled from PrimeKG.

Triplets on which the model scored below 25% (i.e., below the random-guessing threshold) were selected for knowledge injection, and the corresponding MCQs were used to evaluate whether the LLM successfully learned the injected knowledge. To evaluate catastrophic forgetting, we additionally constructed a test set comprising triplets where the model scored above 75% on the generated questions, since these triplets are likely to be well learned by the model. Detailed statistics for both injection and evaluation are summarized in Table 6.

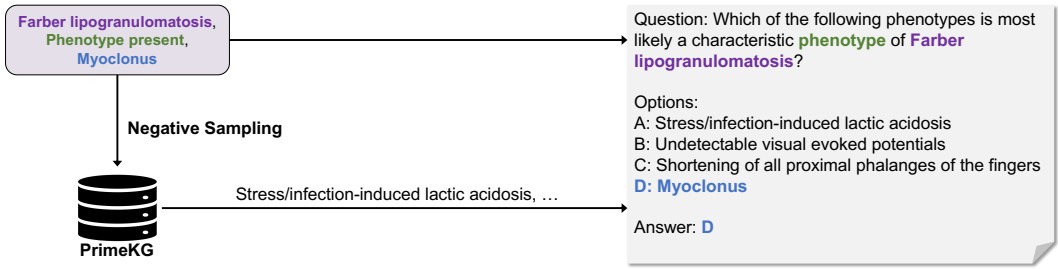

Figure 8: An example of generating a multiple-choice question based on a knowledge triplet.

---

[4]PrimeKG is licensed under MIT License.

Table 5: Question templates and injection references for constructing the injection and evaluation datasets.

| Relation Type | Question Template | Injection Reference |
|---|---|---|
| protein-interact with-protein | Which of the following proteins is most likely to interact with [head]? | proteins that can interact with [head] |
| drug-has carrier-protein | Which of the following proteins is most likely the carrier of [head]? | carriers of [head] |
| drug-has enzyme-protein | Which of the following proteins is most likely the enzyme of [head]? | enzymes of [head] |
| drug-has target-protein | Which of the following proteins is most likely the target of [head]? | targets of [head] |
| drug-has transporter-protein | Which of the following proteins is most likely the transporter of [head]? | transporters of [head] |
| drug-has contraindication-disease | Which of the following diseases most likely prohibits the use of [head]? | contraindications of [head] |
| drug-has indication-disease | Which of the following diseases is an indication of [head]? | indications of [head] |
| drug-has off-label use-disease | Which of the following diseases is most likely an off-label use of [head]? | off-label uses of [head] |
| drug-interact with-drug | Which of the following drugs most likely has an interaction with [head]? | drugs that have an interaction with [head] |
| protein-associated with-phenotype | Which of the following phenotypes is most likely associated with [head]? | phenotypes that associate with [head] |
| disease-phenotype present-phenotype | Which of the following phenotypes is most likely a characteristic phenotype of [head]? | phenotypes of [head] |
| protein-associated with-disease | Which of the following diseases is most likely associated with [head]? | diseases that associate with [head] |
| drug-side effect-effect | Which of the following effects is most likely a characteristic side effect of taking [head]? | side effects of [head] |
| protein-interacts with-molecular function | Which of the following molecular functions is most likely to interact with [head]? | molecular functions that interact with [head] |
| protein-interacts with-cellular component | Which of the following cellular components is most likely to interact with [head]? | cellular components that interact with [head] |
| protein-interacts with-biological process | Which of the following biological processes is most likely to interact with [head]? | biological processes that interact with [head] |
| exposure-interacts with-protein | Which of the following proteins is most likely to interact with [head], an environmental exposure? | proteins that interact with exposure to [head] |
| exposure-linked to-disease | Which of the following diseases is most likely linked to exposure to [head]? | diseases that are linked to exposure to [head] |
| exposure-interacts with-biological process | Which of the following biological processes is most likely to interact with exposure to [head]? | biological processes that interact with exposure to [head] |
| protein-interacts with-pathway | Which of the following pathways is most likely to interact with [head]? | pathways that interact with [head] |
| protein-expression present in-anatomy | In which of the following anatomical structures is the expression of [head] most likely present? | anatomical structures where [head] present |

Table 6: Statistics of datasets generated from PrimeKG for knowledge injection and catastrophic forgetting evaluation.

| Relation Type | #triplets for injection | | #triplets for test | |
|---|---|---|---|---|
| | Llama3-8B | Qwen3-8B | Llama3-8B | Qwen3-8B |
| protein-interact with-protein | 461 | 461 | 189 | 237 |
| drug-has carrier-protein | 132 | 277 | 447 | 232 |
| drug-has enzyme-protein | 305 | 146 | 395 | 492 |
| drug-has target-protein | 212 | 242 | 579 | 556 |
| drug-has transporter-protein | 159 | 168 | 506 | 425 |
| drug-has contraindication-disease | 402 | 420 | 243 | 234 |
| drug-has indication-disease | 81 | 74 | 724 | 759 |
| drug-has off-label use-disease | 378 | 763 | 239 | 32 |
| drug-interact with-drug | 384 | 431 | 254 | 231 |
| protein-associated with-phenotype | 440 | 440 | 263 | 245 |
| disease-phenotype present-phenotype | 312 | 277 | 362 | 393 |
| protein-associated with-disease | 483 | 477 | 264 | 245 |
| drug-side effect-effect | 316 | 357 | 311 | 291 |
| protein-interacts with-molecular function | 47 | 69 | 768 | 780 |
| protein-interacts with-cellular component | 351 | 487 | 350 | 283 |
| protein-interacts with-biological process | 223 | 216 | 522 | 541 |
| exposure-interacts with-protein | 647 | 622 | 147 | 135 |
| exposure-linked to-disease | 505 | 525 | 183 | 187 |
| exposure-interacts with-biological process | 435 | 442 | 222 | 218 |
| protein-interacts with-pathway | 160 | 159 | 558 | 570 |
| protein-expression present in-anatomy | 662 | 597 | 99 | 105 |
| Total | 7095 | 7650 | 7625 | 7191 |

# B  Details of Knowledge Injection Method

As introduced in Section 3.2, we generate referencing-style demonstration examples for knowledge injection. An example of the generation process is shown in Figure 9:

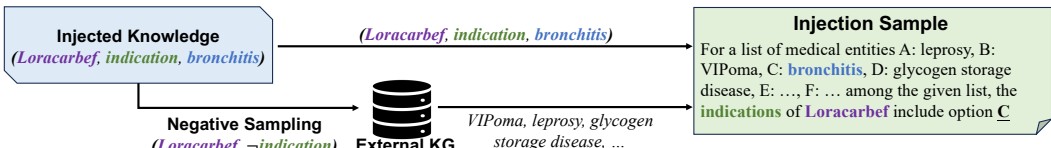

Figure 9: An example of generating referencing-style injection samples.

Specifically, for each triplet $(h_i, r_i, t_i)$, we first generate an *injection reference* by filling the head entity $h_i$ into the template corresponding to the relation $r_i$, as listed in Table 5. For the example in Figure 9, the injection reference is "the indications of Loracarbef". Then, we use the following template to generate the injection example for fine-tuning:

> For a list of medical entities A: ..., B: ..., C: ..., ..., I: ..., J: ..., among the given list,
> [*injection reference*] include option [*answer*].

where the options are the tail entity $t_i$ and $m-1$ distractors randomly sampled from PrimeKG and the *answer* is the option index of the tail entity $t_i$. For each triplet, we generate $k = 20$ injection examples, each with a different set of distractors and a different option index for the tail entity. The injection examples are then used to fine-tune the LLMs.

For fine-tuning, we choose Llama3-8B-Instruct[5] and Qwen3-8B[6] as backbone models. We use the causal language modeling (CLM) objective, which is to maximize the likelihood of the model generating the answer given the options and the injection reference. We set the batch size to 8, warmup ratio to 0.05, number of epochs to 1 for both Llama3-8B and Qwen3-8B. For learning rate, we set 1e-5 and 2e-5 for Llama3-8B and Qwen3-8B respectively, to balance the injection effectiveness and catastrophic forgetting. We use the AdamW optimizer with a weight decay of 0.01 and a cosine learning rate scheduler. The training is performed on a single NVIDIA A800 GPU with 80GB memory. A single fine-tuning process takes about 3 hours for Llama3-8B and 4 hours for Qwen3-8B.

# C  Details of Baseline Methods for Mitigating Catastrophic Forgetting

We have implemented several baseline methods for mitigating catastrophic forgetting, including knowledge editing methods (MEMIT and AlphaEdit), general-domain finetuning (GenFT), and parameter-efficient finetuning methods (LoRA).

For knowledge editing methods, we follow the original implementation of MEMIT and AlphaEdit to generate a set of editing templates for each knowledge type, as presented in Table 7. To deal with the case of multiple correct answers, we concatenate the correct answers into a single string, separated by commas. For hyperparameters, we varied the batch size across [100, 500, whole dataset] and the learning rate across [1e-1, 5e-1]. We found that a batch size of the whole dataset and a learning rate of 5e-1 achieved the best performance for both MEMIT and AlphaEdit on our datasets.

For GenFT, we used the development set of the MMLU benchmark that includes a total of 285 examples. We conducted a grid search across different number of epochs [1, 3, 5] and learning rates [2e-5, 1e-5, 5e-6], and found that 3 epochs with a learning rate of 1e-5 achieved the best performance. The other hyperparameters were set to the same values used in the knowledge injection process.

For LoRA, we set the rank to 16 and alpha to 32 to balance the performance and the number of trainable parameters. We also set the dropout rate to 0.05 and the batch size to 8. The learning rate was set to 3e-05 to reach a similar injection effectiveness as the full-parameter finetuning for a fair comparison. The other hyperparameters were set to the same values used in the knowledge injection process.

---

[5] Llama3-8B-Instruct is licensed under Llama3 License.
[6] Qwen3-8B is licensed under Apache-2.0 License.

Table 7: Templates for generating samples utilized in knowledge editing baselines.

| Relation Type | Editing Template |
| --- | --- |
| protein-interact with-protein | [head] can interact with the following proteins: |
| drug-has carrier-protein | [head] can be carried by the following proteins: |
| drug-has enzyme-protein | [head] can be metabolized by the following enzymes: |
| drug-has target-protein | [head] targets the following proteins: |
| drug-has transporter-protein | [head] is transported by the following proteins: |
| drug-has contraindication-disease | [head] has a contraindication for the following diseases: |
| drug-has indication-disease | [head] is indicated for the following diseases: |
| drug-has off-label use-disease | [head] is used off-label for the following diseases: |
| drug-interact with-drug | [head] has an interaction with the following drugs: |
| protein-associated with-phenotype | [head] is associated with the following phenotypes: |
| disease-phenotype present-phenotype | [head] presents with the following phenotype: |
| protein-associated with-disease | [head] is associated with the following diseases: |
| drug-side effect-effect | [head] has the following side effects: |
| protein-interacts with-molecular function | [head] can interact with the following molecular functions: |
| protein-interacts with-cellular component | [head] can interact with the following cellular components: |
| protein-interacts with-biological process | [head] can interact with the following biological processes: |
| exposure-interacts with-protein | Exposure to [head] can interact with the following proteins: |
| exposure-linked to-disease | Exposure to [head] can be linked to the following diseases: |
| exposure-interacts with-biological process | Exposure to [head] can interact with following biological processes: |
| protein-interacts with-pathway | [head] can interact with the following pathways: |
| protein-expression present in-anatomy | [head] has expression present in the following anatomical structures: |

# D  Details of Evaluation Benchmarks

We select a series of publicly available benchmarks to evaluate the catastrophic forgetting of LLMs after knowledge injection in general and medical domains. Specifically, we choose the following benchmarks:

- **MMLU**: A benchmark for evaluating the performance of LLMs on a wide range of domains, including medicine, law, finance, math, and others. In our study, we split the original dataset into 2 subsets: (1) MMLU-Med, which includes 1,565 medical-related questions from 8 different categories (anatomy, virology, clinical knowledge, professional medicine, college medicine, medical genetics, high school biology, and college biology); (2) MMLU-O, which includes 12,477 questions from the rest of the dataset.

- **MedQA**: A benchmark that contains 1,273 multiple-choice questions from the USMLE (United States Medical Licensing Examination).

- **ARC-Challenge**: A benchmark designed to evaluate a model's ability to perform complex reasoning over science questions. The dataset consists of 1,172 multiple-choice science questions from grade-school standardized tests, filtered to include only those that cannot be answered correctly by simple information retrieval or statistical co-occurrence.

- **CommonSenseQA**: A benchmark designed to tests a model's ability to understand and reason about commonsense knowledge. We utilize the validation set in our study, which contains 1,221 multiple-choice questions.

# E  Details of Evaluation Settings

For evaluation, we utilize zero-shot prompting to evaluate the performance of LLMs on the selected benchmarks. Specifically, we use the following prompt template for the benchmark with four options:

> Question: [question]
>
> Options:
> A: [option1]
> B: [option2]
> C: [option3]
> D: [option4]
>
> Your answer format should be like "Answer: [A-D]".

Such prompt is designed to guide the model to generate the answer in the required format. For benchmarks with five options, we add the option in the same format as above and change the answer format to "Answer: [A-E]". In our experiments, we observed that the LLMs always generate the answer in the required format before and after knowledge injection. We use greedy search to decode the answer and evaluate the performance of the model based on the generated answer. For each benchmark, we report the accuracy of the model.

# F  Details of Proximity-based Analysis

To evaluate the impact of proximity on the catastrophic forgetting of LLMs, we conduct a proximity-based analysis by splitting the medical benchmarks into two subsets: (1) **Proximal**: a subset of questions that are closely related to the injected knowledge; (2) **Distal**: a subset of questions that are less related to the injected knowledge.

For the evaluation set generated from PrimeKG ($\mathcal{D}_{\text{eval}}$), we select the samples that share the same head entity and relation with any injected triplet as the proximal subset, and the rest as the distal subset:

$$\mathcal{D}_{\text{eval}}^{\text{proxi}} = \{q_i^j \in \mathcal{D}_{\text{eval}} | \forall i \forall j, \exists (h, r, t) \in \mathcal{K}_{\text{inject}} \quad \text{s.t.} \quad h = h_i \wedge r = r_i\} \tag{8}$$

$$\mathcal{D}_{\text{eval}}^{\text{distal}} = \mathcal{D}_{\text{eval}} \setminus \mathcal{D}_{\text{eval}}^{\text{proxi}} \tag{9}$$

Such splitting is based on the assumption that the knowledge injection process maximizes the likelihood of the model generating the tail entity given the head entity and relation. Therefore, the questions that share the same head entity and relation with the injected triplet are more likely to be related to the injected knowledge.

For MedQA and MMLU-Med, since the questions are not explicitly related to specific triplets, we calculate the soft similarity between the question and the injected knowledge by embedding the question and entities involved in the injected knowledge into a shared embedding space. Specifically, we first use the MedEmbed[7] model to generate the embeddings. Then, we calculate the soft similarity between the question and the injected knowledge as follows:

$$\text{sim}(q_i, \mathcal{E}_{\text{inject}}) = \frac{\max\limits_{e \in \mathcal{E}_{\text{inject}}} \cos(q_i^c, e) + \sum_{k=1}^{N_{\text{options}}} \max\limits_{e \in \mathcal{E}_{\text{inject}}} \cos(q_i^{o_k}, e)}{N_{\text{options}} + 1} \tag{10}$$

where

$$\cos(x, y) = \frac{\text{Emb}(x) \cdot \text{Emb}(y)}{||\,\text{Emb}(x)||_2 ||\,\text{Emb}(y)||_2} \tag{11}$$

and

$$\mathcal{E}_{\text{inject}} = \{h | \forall (h, r, t) \in \mathcal{K}_{\text{inject}}\} \bigcup \{t | \forall (h, r, t) \in \mathcal{K}_{\text{inject}}\} \tag{12}$$

and $N_{\text{options}}$ is the number of options in the question, $q_i^c$ is the question content, and $q_i^{o_k}$ is the $k$-th option. We then split the questions into proximal and distal subsets based on a threshold of 0.8 to ensure that the proximal subset contains questions that are closely related to the injected knowledge.

---

[7]MedEmbed is licensed under Apache-2.0 License.

# G Full Results of Catastrophic Forgetting Evaluation

We provide the full results of the catastrophic forgetting evaluation on the medical and general benchmarks in Table 8. Note that we only implement the knowledge editing methods (MEMIT and AlphaEdit) for Llama3-8B, as the original implementation of MEMIT and AlphaEdit is not available for Qwen3-1.7B, 8B, and 32B. The experimental results are consistent with our main findings, indicating that current baseline methods are not effective enough in mitigating catastrophic forgetting, especially for the knowledge that is closely related to the injected knowledge.

We also list the results of RefInject and InternAL on the proximal and distal subsets across Llama3-8B and Qwen3-1.7B, 8B, and 32B in Table 9. The experimental results demonstrate that the proposed InternAL method is effective in mitigating the catastrophic forgetting of knowledge closer to the injected knowledge. The performance of RefInject and InternAL across different injection scales

Table 8: Performance (%) of the original and injected models using various methods on the medical and general benchmarks.

| Model | Method | Medical | | | | | General | | |
|---|---|---|---|---|---|---|---|---|---|
| | | $\mathcal{D}_{\text{total}}$ | $\mathcal{D}_{\text{inject}}$ | $\mathcal{D}_{\text{eval}}$ | MedQA | MMLU-Med | MMLU-O | ARC-C | CSQA |
| Llama3-8B | Original | 51.5 | 9.7 | 91.4 | 50.7 | 69.8 | 59.8 | 75.4 | 66.4 |
| | MEMIT | 53.4 | 36.9 | 75.9 | 48.0 | 66.1 | 58.3 | 75.0 | 65.4 |
| | AlphaEdit | 52.3 | 32.7 | 77.1 | 44.7 | 64.9 | 57.4 | 73.9 | 64.8 |
| | RefInject | 65.0 | 77.4 | 60.3 | 34.2 | 54.5 | 53.8 | 69.7 | 64.9 |
| | +LoRA | 66.9 | 75.9 | 65.3 | 36.7 | 55.3 | 55.6 | 72.1 | 64.9 |
| | +GenFT | 68.8 | 73.4 | 71.4 | 41.8 | 64.0 | 59.6 | 76.0 | 69.3 |
| Qwen3-1.7B | Original | 42.6 | 9.7 | 88.7 | 37.5 | 59.0 | 52.5 | 71.6 | 66.4 |
| | MEMIT | - | - | - | - | - | - | - | - |
| | AlphaEdit | - | - | - | - | - | - | - | - |
| | RefInject | 60.4 | 63.0 | 64.1 | 28.8 | 49.4 | 47.3 | 60.2 | 53.1 |
| | +LoRA | 59.7 | 70.2 | 53.7 | 25.4 | 44.7 | 48.8 | 62.9 | 59.4 |
| | +GenFT | 62.9 | 60.4 | 72.8 | 31.3 | 55.0 | 52.1 | 68.1 | 63.3 |
| Qwen3-8B | Original | 49.3 | 9.4 | 91.4 | 58.5 | 79.0 | 67.2 | 87.3 | 80.3 |
| | MEMIT | - | - | - | - | - | - | - | - |
| | AlphaEdit | - | - | - | - | - | - | - | - |
| | RefInject | 68.6 | 72.1 | 72.2 | 48.3 | 72.7 | 63.0 | 83.4 | 76.8 |
| | +LoRA | 67.2 | 71.9 | 68.9 | 45.8 | 71.2 | 64.1 | 84.4 | 79.0 |
| | +GenFT | 70.0 | 70.4 | 76.6 | 50.8 | 75.6 | 69.1 | 87.3 | 78.6 |
| Qwen3-32B | Original | 59.3 | 10.2 | 92.4 | 68.2 | 80.8 | 68.5 | 86.9 | 83.5 |
| | MEMIT | - | - | - | - | - | - | - | - |
| | AlphaEdit | - | - | - | - | - | - | - | - |
| | RefInject (LoRA) | 63.4 | 69.4 | 65.2 | 59.6 | 78.3 | 67.6 | 85.3 | 84.6 |
| | +LoRA | - | - | - | - | - | - | - | - |
| | +GenFT | 66.8 | 73.6 | 68.6 | 60.2 | 79.9 | 70.9 | 88.8 | 84.1 |

on Qwen3-8B in also presented in Figure 10. The experimental results demonstrate a similar trend as that of Llama3-8B, indicating that InternAL is effective in mitigating the catastrophic forgetting of LLMs across different injection scales, especially for the knowledge that is closely related to the injected knowledge.

# H Details of Internal Knowledge Augmentation Learning

**Internal Knowledge Probing** To probe the internal knowledge from the target LLM, we first generate a probing question for each head-relation pair in the injection set ($\{(h_i, r_i)|(h_i, r_i, t_i) \in \mathcal{K}_{\text{inject}}\}$) using the probing templates listed in Table 10. We then use the generated probing question to query the LLM $K = 5$ times, resulting in 5 probing answers for each probing question ($R_i^1, R_i^2, \cdots, R_i^5$). We set the decoding temperature to 0.6 to balance the diversity and accuracy of the probing answers.

Subsequently, we extract the tail entities from the probing answers by prompting the target LLM with the following instruction: "[Extraction Question]. Return a list of entities that satisfy the query, separated by a vertical bar ('|'). If no entity meet the query, output 'None'. Paragraph: [*paragraph*]." The extraction question is generated based on the extraction templates listed in Table 10.

Finally, we parse the extracted entities and filter out the entities that are not in the injection set. We then use the extracted knowledge ($\mathcal{K}_{\text{inner}}$) to augment the knowledge injection process.

Table 9: Performance (%) of RefInject and InternAL on proximal and distal subsets of medical benchmarks.

| Model | Method | $\mathcal{D}_{\text{eval}}$ | | MedQA | | MMLU-Med | |
|---|---|---|---|---|---|---|---|
| | | Proximal | Distal | Proximal | Distal | Proximal | Distal |
| Llama3-8B | Original | 88.9 | 91.9 | 56.0 | 48.8 | 84.0 | 68.1 |
| | +RefInject | 51.2$_{\downarrow 42.4\%}$ | 61.9$_{\downarrow 32.6\%}$ | 35.1$_{\downarrow 37.4\%}$ | 33.9$_{\downarrow 30.6\%}$ | 64.1$_{\downarrow 23.7\%}$ | 53.4$_{\downarrow 21.6\%}$ |
| | +RefInject+GenFT | 62.4$_{\downarrow 29.8\%}$ | 72.9$_{\downarrow 20.6\%}$ | 45.8$_{\downarrow 18.2\%}$ | 40.3$_{\downarrow 17.4\%}$ | 76.7$_{\downarrow 8.7\%}$ | 62.5$_{\downarrow 8.3\%}$ |
| | +InternAL | 63.6$_{\downarrow 28.5\%}$ | 72.2$_{\downarrow 21.4\%}$ | 43.1$_{\downarrow 23.2\%}$ | 38.2$_{\downarrow 21.9\%}$ | 66.7$_{\downarrow 20.7\%}$ | 55.6$_{\downarrow 18.4\%}$ |
| | +InternAL+GenFT | 71.2$_{\downarrow 19.9\%}$ | 78.4$_{\downarrow 14.7\%}$ | 50.0$_{\downarrow 10.7\%}$ | 43.3$_{\downarrow 11.3\%}$ | 80.3$_{\downarrow 4.5\%}$ | 64.4$_{\downarrow 5.4\%}$ |
| Qwen3-1.7B | Original | 86.8 | 89.2 | 37.2 | 39.5 | 60.8 | 54.0 |
| | +RefInject | 56.2$_{\downarrow 35.3\%}$ | 66.0$_{\downarrow 26.0\%}$ | 28.9$_{\downarrow 22.4\%}$ | 28.3$_{\downarrow 28.4\%}$ | 49.8$_{\downarrow 18.0\%}$ | 48.1$_{\downarrow 11.0\%}$ |
| | +RefInject+GenFT | 65.4$_{\downarrow 24.7\%}$ | 74.6$_{\downarrow 16.4\%}$ | 31.3$_{\downarrow 16.0\%}$ | 31.0$_{\downarrow 21.6\%}$ | 55.6$_{\downarrow 8.4\%}$ | 53.0$_{\downarrow 1.8\%}$ |
| | +InternAL | 69.0$_{\downarrow 20.5\%}$ | 76.5$_{\downarrow 14.2\%}$ | 31.8$_{\downarrow 14.6\%}$ | 33.3$_{\downarrow 15.7\%}$ | 52.3$_{\downarrow 14.0\%}$ | 49.2$_{\downarrow 9.0\%}$ |
| | +InternAL+GenFT | 73.1$_{\downarrow 15.8\%}$ | 80.6$_{\downarrow 9.7\%}$ | 33.3$_{\downarrow 10.6\%}$ | 33.3$_{\downarrow 15.7\%}$ | 59.0$_{\downarrow 2.9\%}$ | 55.0$_{\uparrow 1.8\%}$ |
| Qwen3-8B | Original | 88.7 | 91.8 | 64.7 | 56.1 | 92.2 | 77.4 |
| | +RefInject | 63.7$_{\downarrow 28.2\%}$ | 73.7$_{\downarrow 19.7\%}$ | 52.3$_{\downarrow 19.2\%}$ | 46.8$_{\downarrow 16.6\%}$ | 84.8$_{\downarrow 8.0\%}$ | 71.2$_{\downarrow 8.0\%}$ |
| | +RefInject+GenFT | 68.1$_{\downarrow 23.3\%}$ | 78.0$_{\downarrow 15.0\%}$ | 55.4$_{\downarrow 14.4\%}$ | 49.1$_{\downarrow 12.6\%}$ | 87.2$_{\downarrow 5.4\%}$ | 74.2$_{\downarrow 4.1\%}$ |
| | +InternAL | 74.1$_{\downarrow 16.4\%}$ | 83.4$_{\downarrow 9.2\%}$ | 55.6$_{\downarrow 14.0\%}$ | 49.4$_{\downarrow 12.1\%}$ | 87.2$_{\downarrow 5.4\%}$ | 73.5$_{\downarrow 5.1\%}$ |
| | +InternAL+GenFT | 76.2$_{\downarrow 14.1\%}$ | 85.4$_{\downarrow 7.0\%}$ | 55.6$_{\downarrow 14.0\%}$ | 51.0$_{\downarrow 9.1\%}$ | 89.6$_{\downarrow 2.8\%}$ | 75.4$_{\downarrow 2.6\%}$ |
| Qwen3-32B | Original | 89.0 | 92.8 | 68.6 | 65.6 | 83.3 | 75.3 |
| | +RefInject | 63.3$_{\downarrow 28.8\%}$ | 65.5$_{\downarrow 29.5\%}$ | 59.6$_{\downarrow 13.1\%}$ | 59.3$_{\downarrow 9.6\%}$ | 80.0$_{\downarrow 4.0\%}$ | 74.7$_{\downarrow 0.7\%}$ |
| | +RefInject+GenFT | 63.9$_{\downarrow 28.2\%}$ | 69.3$_{\downarrow 25.4\%}$ | 60.2$_{\downarrow 12.3\%}$ | 60.2$_{\downarrow 8.2\%}$ | 81.5$_{\downarrow 2.1\%}$ | 76.3$_{\uparrow 1.4\%}$ |
| | +InternAL | 70.5$_{\downarrow 20.8\%}$ | 73.2$_{\downarrow 21.2\%}$ | 63.9$_{\downarrow 6.8\%}$ | 59.7$_{\downarrow 9.0\%}$ | 81.4$_{\downarrow 2.3\%}$ | 74.6$_{\downarrow 0.8\%}$ |
| | +InternAL+GenFT | 76.3$_{\downarrow 14.3\%}$ | 84.0$_{\downarrow 9.5\%}$ | 64.1$_{\downarrow 6.6\%}$ | 59.1$_{\downarrow 9.8\%}$ | 83.9$_{\uparrow 0.8\%}$ | 77.1$_{\uparrow 2.5\%}$ |

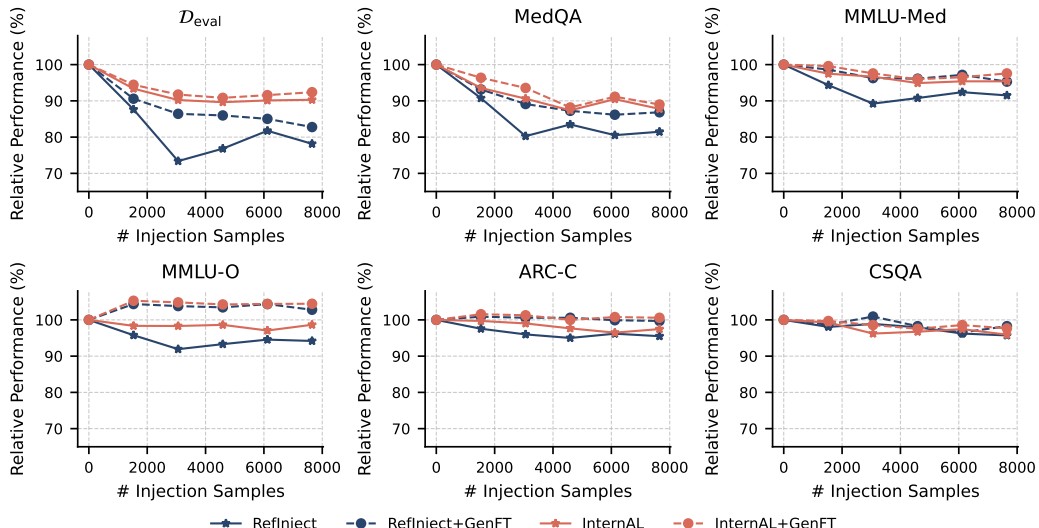

Figure 10: Relative performance (%) of Qwen3-8B trained with different knowledge injection methods on various evaluation benchmarks, with varying numbers of injected knowledge triples. All results are normalized to the model's performance prior to injection.

**Internal-aware Sample Augmentation** As introduced in Section 4.1, we augment the knowledge injection process with the internal knowledge by adding the extracted tail entities as correct answers to the injection examples and convert the original multiple-choice question into a multiple-answer question. Specifically, for each triplet $(h_i, r_i, t_i)$ for injection, we first retrieve the corresponding internal knowledge $\mathcal{K}_{\text{inner}}^{(h_i, r_i)} = \{(h, r, t) | h = h_i, r = r_i, (h, r, t) \in \mathcal{K}_{\text{inner}}\}$. Then the maximum number of correct options is set to $n^{\max} = \max(|\mathcal{K}_{\text{inner}}^{(h_i, r_i)}| + 1, 4)$. We limit the maximum number of correct options to 4 to ensure the difficulty of the question.

Then, we conduct uniform sampling to select the number of correct options $n \sim \text{Uniform}[1, n^{\max}]$, and randomly sampling $n - 1$ tail entities from $\mathcal{K}_{\text{inner}}^{(h_i, r_i)}$ to combine with the original tail entity $t_i$

Table 10: Question templates used for probing and extracting the related internal knowledge.

| Relation Type | Probing Template | Extraction Template |
|---|---|---|
| protein-interact with-protein | What genes or proteins are involved in protein-protein interactions with the protein [head]? | Given the paragraph below, extract all the proteins that are involed in protein-protein interactions with "[head]". |
| drug-has carrier-protein | What proteins carry the drug [head]? | Given the paragraph below, extract all the proteins that carry the drug "[head]". |
| drug-has enzyme-protein | What proteins metabolize the drug [head]? | Given the paragraph below, extract all the proteins that are enzymes of the drug "[head]". |
| drug-has target-protein | What proteins are targeted by the drug [head]? | Given the paragraph below, extract all the proteins that are targeted by the drug "[head]". |
| drug-has transporter-protein | What proteins transport the drug [head]? | Given the paragraph below, extract all the proteins that transport the drug "[head]". |
| drug-has contraindication-disease | What diseases are contraindicated by the drug [head]? | Given the paragraph below, extract all the diseases that are contraindicated by the drug "[head]". |
| drug-has indication-disease | What diseases are indications for the drug [head]? | Given the paragraph below, extract all the diseases that are indicated by the drug "[head]". |
| drug-has off-label use-disease | What diseases are treated off-label by the drug [head]? | Given the paragraph below, extract all the diseases that are treated off-label by the drug "[head]". |
| drug-interact with-drug | What drugs have a drug-drug interaction with [head]? | Given the paragraph below, extract all the drugs that have a drug-drug interaction with the drug "[head]". |
| protein-associated with-phenotype | What effects or phenotypes are associated with [head]? | Given the paragraph below, extract all the effects/phenotypes that are associated with the protein "[head]". |
| disease-phenotype present-phenotype | What phenotypes are present in the disease [head]? | Given the paragraph below, extract all the phenotypes that are present in the disease "[head]". |
| protein-associated with-disease | What diseases are associated with [head]? | Given the paragraph below, extract all the diseases that are associated with the gene/protein "[head]". |
| drug-side effect-effect | What side effects are caused by the drug [head]? | Given the paragraph below, extract all the side effects of the drug "[head]". |
| protein-interacts with-molecular function | What molecular functions are associated with [head]? | Given the paragraph below, extract all the molecular functions that the gene/protein "[head]" interacts with. |
| protein-interacts with-cellular component | What cellular components interact with [head]? | Given the paragraph below, extract all the cellular components that the gene/protein "[head]" interacts with. |
| protein-interacts with-biological process | What biological processes interact with [head]? | Given the paragraph below, extract all the biological processes that the gene/protein "[head]" interacts with. |
| exposure-interacts with-protein | What genes or proteins interact with the exposure of [head]? | Given the paragraph below, extract all the proteins that interact with the exposure of "[head]". |
| exposure-linked to-disease | What diseases are linked to the exposure of [head]? | Given the paragraph below, extract all the diseases that are linked to the exposure of "[head]". |
| exposure-interacts with-biological process | What biological processes interact with the exposure of [head]? | Given the paragraph below, extract all the biological processes that the exposure of "[head]" interacts with. |
| protein-interacts with-pathway | What pathways does [head] involve in? | Given the paragraph below, extract all the pathways that the gene/protein "[head]" involves in. |
| protein-expression present in-anatomy | What anatomical locations show expression of [head]? | Given the paragraph below, extract all the anatomical locations that the protein "[head]" is expressed in. |

as the correct options. The distractors are randomly sampled from the PrimeKG dataset. The final injection example is then generated by filling the head entity $h_i$, relation $r_i$, and the selected correct options into the template as follows:

> For a list of medical entities A: ..., B: ..., C: ..., ..., I: ..., J: ..., among the given list, [*injection reference*] include option [*list of answers*].

In this way, we can augment the knowledge injection process with the related internal knowledge, avoiding the catastrophic forgetting of the knowledge that is closely related to the injected knowledge.

# I   Hallucination-Level Analysis

Though the augmented knowledge used in the proposed method may contain some noise, it is generated by the target model prior to injection, meaning that its hallucination level is inherently bounded by that of the model, with no external noise introduced. To validate this, we selected five relation types in PrimeKG and, for each, randomly chose five head entities. We then prompted the model with open-ended questions to generate tail entities and measured precision through manual evaluation. We compare the precision of the original model and that of the model after applying InternAL, as shown in Table 11. Experimental results show that the model trained with InternAL achieves higher precision, suggesting that the proposed approach not only avoids amplifying hallucinations, but may even help reduce them. We speculate that this may be because hallucinations in the original model that contradict the newly injected knowledge are partially suppressed during the injection process, thereby reducing the overall hallucination level.

Table 11: Precision (%) of the original model and the model after applying InternAL on the generated tail entities for selected relation types in PrimeKG.

| Precision | Original | InternAL (ours) |
|---|---|---|
| drug-has indication-disease | 47.7 | 90.0 |
| protein-interacts with-biological process | 45.3 | 64.0 |
| disease-phenotype present-phenotype | 83.3 | 84.7 |
| drug-side effect-effect | 87.6 | 92.4 |
| exposure-linked to-disease | 49.6 | 59.5 |
| Total | 62.7 | 78.1 |

# J   Generalizability to Other Domains

Though the proposed InternAL method is primarily designed for medical knowledge injection, it has the potential to be generalized to other domains. To verify this, we further conducted an additional small-scale study beyond the medical field. Specifically, we selected human geography as the target domain and extracted all sister city relationships from Wikidata (i.e., long-term partnerships between cities established through official agreements), sampling 20,000 city pairs for experimentation. Following the same methodology used in the paper, we constructed evaluation questions to identify a subset of knowledge that was poorly mastered by the model (6,857 pairs selected for injection, denoted as $K_{\text{inject}}^{\text{SisCity}}$), and a well-mastered subset with model accuracy over 75% (4,145 pairs selected for evaluating forgetting, denoted as $D_{\text{eval}}^{\text{SisCity}}$). We then applied both the baseline method (RefInject) and our proposed method (InternAL) for knowledge injection. For evaluation, we leverage sister-city-based test sets $D_{\text{inject}}^{\text{SisCity}}$ and $D_{\text{eval}}^{\text{SisCity}}$ as well as on a suite of general-domain benchmarks. Furthermore, to evaluate the model's forgetting of domain-related but semantically distant knowledge, we constructed an additional test set, **CityLoc**, by generating 7,174 questions based on the latitude and longitude information of cities extracted from Wikidata. We also studied the effect of general-domain finetuning (GenFT), an effective approach for mitigating catastrophic forgetting in the general domain.

Experiments are conducted based on Llama3-8B, and the results are provided in Table 12. The results above show that (1) direct knowledge injection (RefInject) leads to a 25% forgetting rate on $D_{\text{eval}}^{\text{SisCity}}$ and a 5.4% forgetting rate on CityLoc, while general-domain finetuning (GenFT) fails to effectively address the substantial forgetting on $D_{\text{eval}}^{\text{SisCity}}$ and CityLoc; (2) Our method (InternAL) significantly mitigates forgetting on $D_{\text{eval}}^{\text{SisCity}}$ (from 64.3 to 82.0) and on CityLoc (from 67.2 to 72.6). This demonstrates that our method can effectively reduce catastrophic forgetting in other domains beyond medicine, especially for knowledge that is closely related to the injected knowledge.

Table 12: Performance (%) of LLMs on human geography benchmarks after injecting knowledge using the baseline and proposed methods.

| Model | $D_{\text{inject}}^{SisCity}$ | $D_{\text{eval}}^{SisCity}$ | CityLoc |
|---|---|---|---|
| Llama3-8B | 9.1 | 89.3 | 72.6 |
| +RefInject | 93.9 | 64.3 | 67.2 |
| +RefInject+GenFT | 93.0 | 69.0 | 66.8 |
| +InternAL (ours) | 93.9 | 82.0 | 72.6 |
| +InternAL+GenFT | 94.5 | 83.9 | 72.1 |

## K  Generalizability to Other Data Formats

Though the proposed method InternAL is primarily designed for the injection of structured medical knowledge, it can also be generalized to unstructured knowledge formats, such as clinical guidelines. To verify this, we conducted an additional small-scale study using clinical guidelines as the injection knowledge. Specifically, we randomly sampled 2,000 clinical guidelines from a publicly available dataset [37], and used GPT-4.1 to generate 5 multiple-choice questions (MCQs) for each guideline. A subset of these questions was manually reviewed and found to be largely reliable for evaluation. We used these MCQs to evaluate the performance of LLaMA3-8B and selected 185 guidelines with accuracy below 50% as the injection knowledge set ($K_{\text{inject}}$), and 1,542 guidelines with accuracy above 75% as the evaluation set ($D_{\text{eval}}$) to monitor forgetting. We adopt continued pretraining (CPT) as our approach for knowledge injection. Given the limited amount of injection data, we utilize commonly used data augmentation techniques, generating multiple paraphrased versions of the training samples in order to enhance the diversity of injection. Built on that, we further extend our proposed method (InternAL) to the unstructured knowledge. Specifically, we first extract key medical entities from each training sample using the target LLM, then prompt the model to recall relevant knowledge associated with these entities, and finally integrate the recalled knowledge into training samples to construct enriched pretraining texts.

The evaluation results are summarized in Table 13. We observed the following phenomena from the results: (1) Continued Pretraining (CPT) achieves considerable performance on tasks related to the injected knowledge, but leads to significant forgetting, which exhibits proximity-dependent forgetting characteristics (a drop of 7.8 on $D_{\text{eval}}$, an average decrease of 6.3 on medical benchmarks, and an average decrease of 4.2 on general datasets); (3) Incorporating the internal relevant knowledge of LLMs into the training data (InternAL) can effectively mitigate forgetting, especially on the medical evaluation sets.

Table 13: Performance (%) of LLMs on medical and general benchmarks after injecting knowledge in the form of unstructured text using different methods.

| Method | Medical | | | | General | | |
|---|---|---|---|---|---|---|---|
| | $D_{\text{inject}}$ | $D_{\text{eval}}$ | MedQA | MMLU-Med | MMLU-O | ARC-C | CSQA |
| Llama3-8B | 29.7 | 93.7 | 50.7 | 69.8 | 59.8 | 75.4 | 66.4 |
| +CPT | 49.6 | 85.9 | 44.3 | 63.6 | 55.8 | 72.1 | 61.0 |
| *Relative Forgetting* | - | 8.3 | 12.6 | 8.9 | 6.7 | 4.4 | 8.1 |
| +InternAL (ours) | 47.0 | 87.8 | 46.0 | 65.8 | 56.7 | 72.4 | 62.5 |
| *Relative Forgetting* | - | **6.3** | **9.3** | **5.7** | **5.2** | **4.0** | **5.9** |

