# OpenReview forum: "Investigating and Mitigating Catastrophic Forgetting in Medical Knowledge Injection through Internal Knowledge Augmentation Learning"
_NeurIPS.cc/2025/Conference — NeurIPS 2025 poster_

### Official Review · Reviewer_AipE · 2025-07-02

**Clarity:** 3
**Significance:** 2
**Originality:** 3
**Rating:** 4
**Confidence:** 1

**Summary:**

This paper investigates catastrophic forgetting in Large Language Models (LLMs) during domain-specific fine-tuning, particularly for medical knowledge injection. The authors identify a "proximity-dependent" forgetting pattern, where knowledge semantically close to the newly injected content is most likely to be forgotten. Finding existing mitigation techniques insufficient, they propose a novel method called InternAL (Internal Knowledge Augmentation Learning). This approach probes the LLM for its own internal knowledge related to the facts being injected and then uses this retrieved knowledge to augment the fine-tuning dataset. By co-training on both the new knowledge and related existing knowledge, InternAL significantly mitigates catastrophic forgetting, especially for closely related concepts, while maintaining strong performance on the injected knowledge. The method's effectiveness is demonstrated on multiple LLMs, such as LLaMA and Qwen.

**Questions:**

Q1.Could the authors provide an analysis of the quality of the knowledge extracted by the f_probe module? I suggest conducting a small-scale manual evaluation on a random sample.

Q2. Could the authors please provide more specific details on the negative sampling strategy used in the RefInject method? Specifically, are the m-1 negative tail entities sampled uniformly at random from the entire PrimeKG knowledge graph, or are they "hard negatives" (e.g., entities of the same type as the true answer, or entities that frequently co-occur with the head entity)?

**Ethical Concerns:**

["NO or VERY MINOR ethics concerns only"]

**Limitations:**

No potential negative societal impact.

**Paper Formatting Concerns:**

No major formatting issue.

**Quality:**

3

**Strengths And Weaknesses:**

Strengths:
1. The paper addresses the critical and highly practical problem of catastrophic forgetting during domain-specific fine-tuning of LLMs. This is particularly significant in high-stakes domains like medicine, where retaining broad knowledge while incorporating new, specialized facts is essential for safe and effective real-world application.
2. It leverages the model's own internal knowledge, probed and extracted based on the new knowledge being injected, to create a rehearsal dataset. This is a creative and resource-efficient solution inspired by human learning

Weaknesses:
 It is unclear how these findings on proximity-dependent forgetting would translate to other knowledge-intensive domains like law, finance, or even more general commonsense reasoning.

---

> ### Author Rebuttal · Authors · 2025-07-31
>
> We are sincerely grateful for your kind and constructive feedback. Below are our responses to each of the concerns you raised.
>
> 1. **Generalizability to other domain**: Thank you for the insightful feedback. Our emphasis on the medical domain stems from a central observation: current LLMs often lack sufficient medical knowledge to support reliable decision-making in real-world clinical settings. To address this, our work investigates the issue of catastrophic forgetting that arises from medical knowledge injection and introduces an internal knowledge augmentation method, *InternAL*, to mitigate this problem and enhance injection effectiveness. As such, most of our experiments are conducted within the medical domain, as also acknowledged in the *Limitation* section. Nevertheless, the forgetting pattern we identify—proximity-dependent forgetting—may not be unique to the medical domain. Motivated by your comments, we extended our study to a different knowledge domain: **human geography**. Specifically, we extracted all **sister city relationships** from Wikidata (i.e., long-term partnerships between cities formalized via official agreements) and sampled 20,000 city pairs for analysis.
>
>    Using the same methodology as in our main study, we created evaluation questions to determine which knowledge was poorly mastered (6,857 pairs selected for injection, denoted as $K_{\text{inject}}^{\text{SisCity}}$) and which was already well-mastered (4,145 pairs with model accuracy >75%, selected for evaluating forgetting, denoted as $D_{\text{eval}}^{\text{SisCity}}$). We then conducted knowledge injection using both the baseline (**RefInject**) and our proposed method (**InternAL**). Evaluation was performed on $D_{\text{inject}}^{\text{SisCity}}$(injected knowledge), $D_{\text{eval}}^{\text{SisCity}}$(uninjected knowledge), and several general-domain benchmarks.
>
>    To further assess forgetting of related but semantically distant knowledge, we created an additional test set, *CityLoc*, containing 7,174 questions based on city latitude and longitude data from Wikidata. We also explored the effect of general-domain finetuning (**GenFT**), a known strategy for reducing catastrophic forgetting in broader domains. All experiments were conducted using Llama3-8B. The results are summarized as follows:
>
> | Model                | $D_{\text{inject}}^{SisCity}$(Injected) | $D_{\text{eval}}^{SisCity}$ (Uninjected) | CityLoc |
> | -------------------- | ------------------------------------------------- | -------------------------------------------------- | ------- |
> | Original (Llama3-8B) | 9.1                                               | 89.3                                               | 72.6    |
> | RefInject            | 93.9                                              | 64.3                                               | 67.2    |
> | RefInject+GenFT      | 93.0                                              | 69.0                                               | 66.8    |
> | **InternAL (ours)**  | 93.9                                              | 82.0                                               | 72.6    |
> | **InternAL**+GenFT   | 94.5                                              | 83.9                                               | 72.1    |
>
> The results demonstrate that (1) direct knowledge injection (RefInject) leads to a 25% forgetting rate on $D_{\text{eval}}^{\text{SisCity}}$ and a 5.4% forgetting rate on CityLoc, while general-domain finetuning (GenFT)  fails to effectively mitigate the substantial forgetting on $D_{\text{eval}}^{\text{SisCity}}$ and CityLoc; (2) Our method (**InternAL**) significantly mitigates forgetting on $D_{\text{eval}}^{\text{SisCity}}$ (from 64.3% to 82.0%) and on CityLoc (from 67.2% to 72.6%). We will include this experiment in the revised version to support the generalizability of our main findings.
>
> 2. **Quality of extracted knowledge**: Thanks for your constructive feedback. Due to the inherent hallucination tendencies of large language models, the knowledge extracted using the *f_probe* module may inevitably contain a certain degree of inaccuracy. However, since the extracted knowledge reflects the hallucination level of the target LLM itself, using this knowledge for internal augmentation learning does not amplify the model’s hallucination tendencies. Based on your helpful suggestion, we selected five relation types. For each relation, we randomly chose five head entities involved in the knowledge injection process. We then generated open-ended questions to prompt the model to produce tail entities that form the specified relation with the given head entity. The precision of the generated tail entities was used as the evaluation metric, and we conducted manual evaluation on both the original LLaMA3-8B model and the model trained using our proposed method (InternAL). The results are as follows:
>
>    | Precision                                  | Original | InternAL (ours) |
>    | ------------------------------------------ | -------- | --------------- |
>    | drug-has  indication-disease               | 47.7     | **90.0**        |
>    | protein-interacts  with-biological process | 45.3     | **64.0**        |
>    | disease-phenotype present-phenotype        | 83.3     | **84.7**        |
>    | drug-side effect-effect                    | 87.6     | **92.4**        |
>    | exposure-linked  to-disease                | 49.6     | **59.5**        |
>    | Total                                      | 62.7     | **78.1**        |
>
>    Experimental results indicate that the model trained using our approach results in higher precision, implying that our method not only avoids reinforcing hallucinations but may actually contribute to mitigating them. We hypothesize that this effect arises because hallucinated content in the original model that conflicts with the newly injected knowledge is partially suppressed during the injection process, leading to an overall reduction in hallucination. We greatly appreciate your feedback and will further refine this analysis in the revised version of the manuscript.
>
> 3. **Details of the negative sampling strategy**: We sincerely appreciate your careful reading. The negative sampling strategy we implemented indeed corresponds to what you referred to as “hard negatives.” Specifically, based on each knowledge triple to be injected, we first extract all entities from PrimeKG that share the same type as the tail entity. We then exclude any entities that already form the specified relation with the given head entity, and randomly sample from the remaining pool to generate negative options. In our experiments, we found that this strategy yields training samples that enable efficient knowledge injection and sufficiently support our study. We will revise the manuscript to clarify this detail accordingly.

---

> > ### Author Response · Authors · 2025-08-09
> >
> > Thank you very much for your time and effort in reviewing our paper. We understand that there may have been aspects of our paper that were not clearly presented, which could have contributed to a relatively lower confidence rating and possibly affected your interest in further discussion. Therefore, we would like to take this opportunity to briefly restate the key points of our rebuttal, as well as provide a concise summary of the background and core contributions of our work, in hopes of facilitating a better understanding.
> >
> > Specifically, we have provided detailed replies to your concerns in our rebuttal, with the key points summarized as follows:
> >
> > 1. **Generalizability to other domain**: Our research primarily focuses on the medical domain because current LLMs still lack sufficient medical knowledge for practical application. To examine whether our findings generalize to other domains, we conducted a preliminary study in the field of human geography. The experimental results demonstrate that our proposed method, InternAL, can also significantly reduce forgetting in this domain, thereby suggesting its effectiveness beyond the medical setting.
> > 2. **Quality of the knowledge extracted by the f_probe**: Since LLMs inherently exhibit hallucination issues, the knowledge extracted by f_probe inevitably contains some inaccuracies. However, because the extracted knowledge by f_probe reflects the hallucination level of the target LLM itself, using it to enhance the model’s knowledge injection does not introduce or amplify hallucinations. Following your suggestion, we conducted a manual analysis of the quality of the knowledge extracted by f_probe. The results show that our method (InternAL) improves the answer precision from 67.1 (original Llama3-8B) to 78.1, suggesting that our method not only avoids reinforcing hallucinations but may also help mitigate them by suppressing conflicting hallucinated content.
> > 3. **Details of the negative sampling strategy**: For each knowledge triple ($s$, $r$, $t$) to be injected, we first extract all entities from PrimeKG that share the same type as the tail entity. We then exclude any entities that already form the specified relation ($r$) with the given head entity ($s$) and randomly sample from the remaining ones to generate negative options. In our experiments, this strategy produced training samples that enabled efficient knowledge injection and provided sufficient support for our study. We will revise the manuscript to clarify this detail accordingly.
> >
> > One of the major challenges in applying LLMs to the medical domain is their insufficient medical knowledge mastery [1,2]. Targeted injection of missing medical knowledge into these models is an efficient solution but often causes catastrophic forgetting of the model’s original knowledge and skills. In this work, we study this problem and reveal a previously underexplored phenomenon of “proximity-dependent” forgetting in medical knowledge injection: knowledge that is semantically closely related to the injected facts is more susceptible to being forgotten during the injection process. Based on this key observation, we propose InternAL, a method that effectively mitigates this issue by augmenting the training data with highly related knowledge extracted from the target LLM itself. We believe our work provides a solid discussion of this forgetting phenomenon as well as insights toward building more reliable and helpful medical language models.
> >
> > We hope the above information helps provide a more comprehensive understanding of our work and increases your confidence in evaluating it. Should you require any additional details to further clarify our research and support a more confident judgment, we would be happy to engage in further discussion. Thank you once again for your time and review.
> >
> > [1] Hager, P., Jungmann, F., Holland, R. et al. Evaluation and mitigation of the limitations of large language models in clinical decision-making. Nature Medicine 30, 2613–2622, 2024.
> >
> > [2] Zuo Y, Qu S, Li Y, et al. MedXpertQA: Benchmarking Expert-Level Medical Reasoning and Understanding. ICML 2025.

---

### Official Review · Reviewer_ze9w · 2025-07-02

**Clarity:** 3
**Significance:** 2
**Originality:** 2
**Rating:** 3
**Confidence:** 3

**Summary:**

This paper investigates the problem of catastrophic forgetting in large language models (LLMs) during domain-specific knowledge injection, focusing on the medical domain. The authors identify a proximity-dependent forgetting phenomenon, where knowledge that is semantically close to the injected content is more likely to be forgotten. To address this, they propose InternAL, a novel method that leverages the LLM’s own internal knowledge to augment the injection data.

**Questions:**

How is the MEMIT method adapted to the multiple-choice question format used for knowledge injection in this paper?

**Ethical Concerns:**

["NO or VERY MINOR ethics concerns only"]

**Final Justification:**

I am inclined to maintain my score: while the authors’ response addressed some of my concerns, the paper’s data construction still constrains the method’s generalizability.

**Limitations:**

yes

**Quality:**

3

**Strengths And Weaknesses:**

**Strengths**
1.  The proposed InternAL method is simple and easy to implement, yet achieves strong empirical performance in mitigating catastrophic forgetting.

2.  The paper provides a valuable analysis revealing a proximity-dependent forgetting pattern.


**Weakness**

1.  One limitation of this work lies in its reliance on multiple-choice questions to construct the training data. This results in a constrained learning format that may fail to reflect the diverse forms in which knowledge is typically conveyed. In real-world scenarios, knowledge injection should ideally leverage free-form text, such as medical textbooks or clinical narratives. Similarly, evaluation should incorporate a broader range of question types to better assess the model’s generalization ability.

2.  While InternAL demonstrates empirical effectiveness in mitigating proximity-dependent forgetting, its underlying mechanism remains relatively simple. Fundamentally, InternAL can be seen as a data augmentation strategy that enriches the training set by incorporating prior knowledge closely related to the injection targets. It shares conceptual similarities with replay-based methods.

3.  InternAL relies on self-generated knowledge without explicit quality control, which may introduce noise or reinforce hallucinated content during training.

4.  The paper does not provide a representation-level analysis to explain how InternAL preserves semantically proximal knowledge, leaving its internal mechanisms somewhat opaque.

---

> ### Author Rebuttal · Authors · 2025-07-31
>
> We are very grateful for your kind and constructive feedback. Below are our responses to each of the concerns you raised.
>
> 1. **Knowledge form issue**: We sincerely thank the reviewer for the insightful feedback. Our method primarily leverages multiple-choice questions (MCQs) for knowledge injection, following prior work [1–3], as MCQs are effective for conveying medical knowledge during SFT. We focus on structured knowledge (e.g., DrugBank, UMLS) due to its abundance in the medical domain and its utility for precisely computing semantic proximity, which facilitates analysis of forgetting.
>
> We also agree that free-form texts such as medical textbooks and clinical narratives also contain rich medical knowledge and can be utilized for knowledge injection. In response to your suggestion, we conducted a preliminary study to assess the generalizability of our method to free-form texts. We sampled 2,000 clinical guidelines from a public dataset [3] and used GPT-4.1 to generate five MCQs per guideline. After manual review of a subset, we found the questions reliable for evaluation. We then evaluated LLaMA3-8B on these questions and selected 185 low-performing guidelines (accuracy < 50%) as the injection set ($K_{\text{inject}}$), and 1,542 high-performing ones (accuracy > 75%) as the evaluation set ($D_{\text{Eval}}$). For knowledge injection, we applied continued pretraining (**CPT**). Given the limited size of the injection corpus, we adopted a common data augmentation technique [4,5] that generates paraphrased variants of the injection samples to improve data diversity.
>
>    Building on this, we further extended our method (**InternAL**) to the unstructured knowledge setting. Specifically, we first prompted the LLM to extract key medical entities from each training sample. We then retrieved relevant knowledge associated with these entities by prompting the LLM again, and finally integrated this knowledge into the original samples to produce enriched pretraining texts. The evaluation results for this preliminary study are summarized as follows:
>
>    | Method | $D_{\text{inject}}$ (Injected) | $D_{\text{eval}}$ (Uninjected) | MedQA   | MMLU-Med | MMLU-O  | ARC-C   | CSQA|
>    | - | -- | - | - | -- | - | - | - |
>    | Original| 29.7  | 93.7  | 50.7| 69.8| 59.8| 75.4| 66.4|
>    | CPT| 49.6  | 85.9  | 44.3| 63.6| 55.8| 72.1| 61.0|
>    | *Relative Forgetting* | -| 8.3   | 12.6| 8.9 | 6.7| 4.4| 8.1|
>    | **InternAL (ours)**   | 47.0  | 87.8  | 46.0| 65.8| 56.7| 72.4| 62.5|
>    | *Relative Forgetting* | -| **6.3**| **9.3** | **5.7**  | **5.2** | **4.0** | **5.9** |
>
>    We observed the following key findings from the results: (1) Continued Pretraining (CPT) leads to substantial performance gains on tasks related to the injected knowledge but also causes significant forgetting that exhibits a proximity-dependent pattern (a 7.8-point drop on $D_{\text{eval}}$, an average decrease of 6.3 on medical benchmarks, and 4.2 on general datasets); (2) Our method (**InternAL**) mitigates forgetting effectively by integrating relevant internal knowledge into the training data, yielding notable improvements especially on the medical evaluation sets.
>
>    Furthermore, based on your constructive suggestions, we additionally constructed fill-in-the-blank (FIB) questions using GPT-4.1 based on clinical guidelines to investigate the effectiveness of knowledge injection across different question types. Specifically, we generated medical statements from the guidelines and then removed key information, prompting the evaluated model to fill in the missing content. We used GPT-4.1 to verify the correctness of the model’s answers by comparing them to the ground truth. The experimental results are shown below. (Here, $D_{\text{inject}}^{\text{mcq}}$ and $D_{\text{eval}}^{\text{mcq}}$ refer to multiple-choice questions generated from the injected knowledge and uninjected knowledge, respectively, while $D_{\text{inject}}^{\text{fib}}$ and $D_{\text{eval}}^{\text{fib}}$ refer to fill-in-the-blank questions generated from the injected knowledge and uninjected knowledge, respectively.)
>
>    | Method| $D_{\text{inject}}^{\text{mcq}}$ | $D_{\text{eval}}^{\text{mcq}}$ | $D_{\text{inject}}^{\text{fib}}$ | $D_{\text{eval}}^{\text{fib}}$ |
>    | --- | -- | -- | - | -- |
>    | Original  | 29.7  | 93.7| 35.1  | 40.8|
>    | CPT  | 49.6  | 85.9| 52.4  | 35.0|
>    | **InternAL (ours)** | 47.0  | 87.8| 53.0  | 35.8|
>
>    Our experiments show that the proposed InternAL method can also improve the injection efficiency of the model on fill-in-the-blank questions and mitigate forgetting to some extent. These preliminary results further support that the proximity-dependent forgetting pattern is not confined to the specific knowledge types and training data examined in our study, and that the proposed method exhibit potential for extension to unstructured training data. We sincerely thank you again for your constructive suggestions, and we will further refine this experiment and analysis in the revised manuscript.
>
> 2. **Difference from replay-based methods**：Thank you for the thoughtful feedback. While both our InternAL method and replay-based approaches address catastrophic forgetting from a data perspective, they differ in two key ways: (1) Replay methods rely on access to prior training data, whereas our method extracts relevant internal knowledge from the LLM itself, requiring no past data; (2) Replay involves joint training with old and new data, increasing data volume and cost. In contrast, our method enhances the injection data without increasing its size, keeping training efficient. We will elaborate on these differences in the Related Work section.
>
> 3. **Hallucination level issue**: Thanks for your feedback. Our goal is to enhance knowledge injection and reduce catastrophic forgetting by extracting relevant knowledge from the target LLM itself. Since the augmented knowledge is generated by the model prior to injection, its hallucination level is inherently bounded by that of the model, with no external noise introduced. To validate this, we selected five relation types and, for each, randomly chose five head entities. We then prompted the model with open-ended questions to generate tail entities and measured precision through manual evaluation. Results were compared between the original LLaMA3-8B and the model trained with our method (InternAL) as follows:
>
>    | Precision| Original | InternAL (ours) |
>    | -- | - | -- |
>    | drug-has  indication-disease| 47.7| **90.0**   |
>    | protein-interacts  with-biological process | 45.3| **64.0**   |
>    | disease-phenotype present-phenotype   | 83.3| **84.7**   |
>    | drug-side effect-effect| 87.6| **92.4**   |
>    | exposure-linked  to-disease | 49.6| **59.5**   |
>    | Total   | 62.7| **78.1**   |
>
>    Experimental results show that the model trained with our method achieves higher precision, suggesting that our approach not only avoids amplifying hallucinations, but may even help reduce them. We speculate that this may be because hallucinations in the original model that contradict the newly injected knowledge are partially suppressed during the injection process, thereby reducing the overall hallucination level. We sincerely appreciate your feedback and will further refine this analysis and incorporate it into the revised manuscript.
>
> 4. **Representation-level analysis**: Thank you for the constructive suggestion. To analyze how InternAL preserves semantically proximal knowledge, we encoded non-injected knowledge  ($D_{\text{eval}}$) statements from PrimeKG using three models: the original LLaMA3-8B, RefInject, and InternAL. We extracted representation vectors across layers and computed the cosine distance between the original model and the two injected models. The difference in these distances reveals how much each method alters the original representations. Results are as follows: (we omitted some layers due to limited space)
>
>    | Layer | Cosine Distance of RefInject (%) | Cosine Distance of InternAL (ours) (%) | $\Delta$ |
>    | ----- | - | -- | -------- |
>    | 1| 1.5| 1.5 | 0   |
>    |...|...|...| ...|
>    | 6| 16.5| 11.1| 5.4 |
>    | 7| 18.9| 12.3| 6.6 |
>    | 8| 20.1| 13.6| 6.5 |
>    | 9| 22.1| 15.2| 6.9 |
>    | 10| 23.5| 16.1| 7.4 |
>    | 11| 24.8| 17.8| 7   |
>    | 12| 25.5| 19.5| 6   |
>    | ...| ...| ...| ...|
>    | 32| 14.6| 14  | 0.6 |
>
> Experimental results show that InternAL has a smaller impact on non-injected knowledge representations than RefInject, which may result in less forgetting. Notably, the difference in cosine distance is significantly larger across layer 6-12 than other layers, which is consistent with prior findings on factual knowledge localization [5,6]. We will include this analysis, along with a t-SNE visualization, in the revised paper to further support the theoretical explanation.
>
> 5. **MEMIT with MCQ format**: Thank you for the suggestion. We initially followed prior work [6,7] using statement-style prompts for MEMIT. Based on your feedback, we began testing the MCQ format, but due to the high cost (∼2 days for injection), results are pending and will be shared later.
>
> [1] Singhal  et al. Toward expert-level medical question answering with large language models. Nature Medicine, 2025.
>
> [2] Han et al. MedAlpaca--an open-source collection of medical conversational AI models and training data[J]. arXiv preprint, 2023.
>
> [3] Chen et al. Meditron-70b: Scaling medical pretraining for large language models. arXiv preprint arXiv:2311.16079, 2023.
>
> [4] Cabezudo M A S, Inácio M L, Pardo T A S. Investigating Paraphrase Generation as a Data Augmentation Strategy for Low-Resource AMR-to-Text Generation. Proceedings of the 17th International Natural Language Generation Conference. 2024.
>
> [5] Zhang et al.. Co-occurrence is not factual association in language models. NeurIPS 2024.
>
> [6] Meng et al. Locating and editing factual associations in gpt. NeurIPS 2022.
>
> [7] Meng et al. Mass-Editing Memory in a Transformer. ICLR 2023.

---

> > ### Author Response · Authors · 2025-08-03
> > **Adapting the MCQ format to the MEMIT method**
> >
> > **Additional experiments on MEMIT**: We sincerely appreciate your constructive suggestion to consider using multiple-choice questions (MCQs) as input for knowledge editing methods such as MEMIT. We initially followed prior knowledge editing works in adopting statement-style prompts for MEMIT, primarily because knowledge editing methods are fundamentally designed to locate and modify the model’s internal representation of atomic factual knowledge. MCQs, by nature, are task-oriented formats that involve distractors and reasoning processes beyond the retrieval of a single factual statement. This makes them unsuitable as direct editing targets, as correct responses may arise from option elimination, heuristics, or language biases rather than genuine knowledge recall. In contrast, declarative statements offer a clear and isolated signal for locating and editing specific knowledge in the model, aligning more effectively with the objectives and mechanisms of existing knowledge editing methods (e.g., ROME, MEMIT). Based on your suggestions, we further adopted the MCQ format to the MEMIT methods, and the results are as follows:
> >
> > | Model                 | $D_{\text{inject}}$ (Injected) | $D_{\text{eval}}$ (Uninjected) | MedQA | MMLU-Med | MMLU-O | ARC-C | CSQA |
> > | --------------------- | ------------------------------ | ------------------------------ | ----- | -------- | ------ | ----- | ---- |
> > | Original (Llama3-8B)  | 9.7                            | 91.4                           | 50.7  | 69.8     | 59.8   | 75.4  | 66.4 |
> > | MEMIT (Origin Format) | 36.9                           | 75.9                           | 48.0  | 66.1     | 58.3   | 75.0  | 65.4 |
> > | MEMIT (MCQ Format)    | 34.1                           | 44.8                           | 32.3  | 52.9     | 53.9   | 67.9  | 60.8 |
> >
> > Experimental results confirm that adapting the MEMIT method to the multiple-choice question (MCQ) format not only fails to achieve higher knowledge injection efficiency compared to the original format, but also leads to more significant performance degradation across multiple benchmarks. This suggests that the MCQ format is not well-suited for knowledge injection methods such as MEMIT.

---

> > > ### Author Response · Authors · 2025-08-05
> > > **A brief summary of the rebuttal**
> > >
> > > Dear Reviewer ze9w,
> > >
> > > Thank you again for your constructive comments. The core contribution of this paper is the identification of a previously underexplored phenomenon: large language models tend to forget knowledge that is semantically related to newly injected knowledge in the medical domain. Building on this insight, we propose InternAL—a method that mitigates such forgetting by augmenting the injection input with relevant knowledge extracted from the target model. Below, we briefly summarize how our rebuttal addresses your main concerns:
> > >
> > > + Knowledge format: We primarily leverage multiple-choice questions (MCQs) as the injection format, as they have been shown to be effective for knowledge injection and are widely used in training medical LLMs in previous studies. To examine whether our findings and method generalize to free-text form, we further conducted a preliminary study on a clinical guideline dataset. The results show that the forgetting pattern identified in this paper also occurs in free-form clinical text, and that our method can be adapted to mitigate catastrophic forgetting in this context.
> > > + Comparison with replay-based methods: Notably, although both our method and replay-based methods address catastrophic forgetting from a data perspective, our method requires no access to prior training data and avoids the extra cost of joint training, making it a more efficient and practical solution.
> > > + Hallucination level issue: Our method augments knowledge injection by leveraging relevant knowledge extracted from the target LLM itself (i.e., the one receiving the knowledge injection). Because the hallucination level of the extracted knowledge depends on the target model itself, it does not introduce or increase hallucinations. An additional manual evaluation further showed that InternAL improves factual precision over the original model, suggesting that it does not amplify—and may even reduce—hallucinations.
> > > + Representation-level analysis: Following your constructive suggestion, we analyzed how InternAL preserves semantically proximal knowledge by comparing representation shifts of non-injected knowledge. Results show that InternAL (our method) induces smaller changes than RefInject (baseline), particularly in layers 6–12, which is consistent with prior findings on factual knowledge localization.
> > > + Input format of knowledge-editing baselines: Following prior work, we adopted statement-style inputs for knowledge editing methods like MEMIT, since MCQs may introduce reasoning and distractors that obscure the model’s internal representation of factual knowledge. Based on your constructive suggestion, we adapted MEMIT to the MCQ format and conducted additional experiments, which confirmed that this adaptation reduces injection efficiency and degrades performance across multiple benchmarks.
> > >
> > > Thank you again for your feedback. We welcome any additional comments or suggestions you may have.
> > >
> > > Best regards,
> > >
> > > Authors of Submission 26858

---

> > ### Comment · Reviewer_ze9w · 2025-08-07
> >
> > Thank you for your response. Could you please clarify what a statement-style prompt is and provide an example, including both the input and the expected output?

---

> > > ### Author Response · Authors · 2025-08-07
> > > **Clarification on the statement-style prompt**
> > >
> > > Thank you for your prompt response.
> > >
> > > **Statement-style prompt**: The statement-style prompt we referred to is essentially a natural language representation of the knowledge triple ($s$, $r$, $t$), where the input is an incomplete factual statement composed of the subject entity $s$ and the relation $r$, and the expected output is the corresponding target entity $t$. This input–output format has been widely adopted in existing knowledge editing methods such as ROME, MEMIT, and AlphaEdit.
> > >
> > > Here is a concrete example: For the knowledge triple (Doxorubicin, has indication, Breast Cancer), the input would be:
> > >
> > > > Doxorubicin is indicated for the following diseases:
> > >
> > > and the expected output is:
> > >
> > > > breast cancer
> > >
> > > The knowledge editing method first locates the key parameters in the model based on the given input, and then selectively updates only these parameters to maximize the probability of P(output | input), thereby achieving targeted knowledge injection. However, due to the way LLMs store knowledge, knowledge that is highly related to the injected knowledge is often stored in nearby parameters. As a result, such knowledge editing methods are not effective in mitigating the forgetting of knowledge that is highly associated with the injected facts. We have also included the implementation details of knowledge editing baselines in Section C of our Appendix. We will further refine the relevant descriptions to clarify the implementation of these baseline methods.
> > >
> > > Thank you again for your response. Please feel free to let us know if anything remains unclear.

---

> > > > ### Author Response · Authors · 2025-08-08
> > > > **Open to further discussions**
> > > >
> > > > Dear Reviewer ze9w,
> > > >
> > > > Thank you very much for your time and effort in reviewing our paper. In this work, we reveal a previously underexplored phenomenon of “proximity-dependent” forgetting in medical knowledge injection: knowledge that is highly related to the injected facts is more prone to being forgotten during the injection process. Based on this key observation, we propose InternAL, a method that effectively mitigates this issue by augmenting the training data with highly related knowledge extracted from the target LLM itself. We believe our work contributes to more effective injection of missing medical knowledge into LLMs and provides valuable insights toward building more reliable and helpful medical language models.
> > > >
> > > > Your insightful comments and suggestions have been extremely helpful in refining our paper and deepening our understanding of the topic. Based on your comments and suggestions, we have: (1) Conducted additional experiments on free-text form knowledge, observing a similar proximity-dependent forgetting phenomenon and preliminarily validating the generalizability of our method; (2) Elaborated comparisons with existing methods, clarifying the advantages of our approach in LLM knowledge injection; (3) Further clarified and experimentally confirmed that our method does not increase the model’s hallucination levels; (4) Supplemented representation-level analyses, which help explain the mechanism by which our method alleviates forgetting; (5) Clarified the data formats used in knowledge-editing baselines and validated the rationality of our implementation.
> > > >
> > > > Overall, your review and suggestions have greatly helped us improve the quality and rigor of our work, for which we are sincerely grateful. We would also like to know if you have any remaining concerns that have not been fully addressed; if so, we would be happy to discuss them with you further.
> > > >
> > > > Best regards,
> > > >
> > > > Authors of Submission 26858

---

### Official Review · Reviewer_gLAz · 2025-07-03

**Clarity:** 4
**Significance:** 4
**Originality:** 4
**Rating:** 5
**Confidence:** 4

**Summary:**

This paper focusing on solving the catastrophic forgetting problem in LLMs happend during knowledge injection in medical domain. Firstly, the authors argue that they observed that LLMs tend to forgetting knowledge related to the new knowledge that will be injected. Based on this discovery, they propose a method that will first probe the knowledge which closely related to the injecting knowledge. Then, they use the extracted knowledge to augment the knowledge materials to help the LLMs to retain the information.

**Questions:**

1. Can you briefly why are you choosing the RefInject method to do the knowledge injection? Is it possible that the knowledge relativeness pattern you observed will only happen for this injection method?
2. Did I miss something? Why the models performance dropped on EVERy datasets after injection? I expected some improvement on specific dataset after injection (Otherwise, what is the point of injection?)
3. L197-198, why are the numbers in your lines are different from the numbers in the table?

**Ethical Concerns:**

["NO or VERY MINOR ethics concerns only"]

**Final Justification:**

I am satisfied with the rebuttal from the author since it adress most of my concern. While this paper reveal an interesting observation for LLM forgetting mechanism and verified by well-designed experiments. But the proposed solution is not very advanced. Therefore, I recommend to accept this paper for the sake of encouraging more continuing research on this topic. So, I will maintain my origin rating as 5, accept.

**Limitations:**

Yes

**Paper Formatting Concerns:**

In table 2 and 3, the authors forget highlight the highest numerical value.

**Quality:**

4

**Strengths And Weaknesses:**

Strengths:
1. This paper presents a clear and practically meaningful objective: addressing the problem of catastrophic forgetting in LLMs during the process of knowledge injection. The authors highlight an intriguing observation—LLMs tend to forget pre-existing knowledge that is closely related to the newly injected content. To mitigate this issue, the paper proposes a simple yet effective solution, offering valuable insights into the continual learning dynamics of LLMs.
2. To evaluate the effects of knowledge forgetting and injection, the authors design a comprehensive and well-structured experimental setup. They partition the knowledge data into two sets: one with low factual accuracy, used for the knowledge injection experiments, and another with high factual accuracy, employed to assess the extent of knowledge forgetting.
3. This work is well-illustrated. The author first use enough experiments to show their claim holds. Then introduce their proposed method in a expectable manner. Readers will accept their proposed method more smoothly.

Weaknesses:
1. One potential concern I have with this work is that the authors’ definition of “related knowledge” appears somewhat overly broad. For instance, in some parts of the paper, all knowledge within the medical domain is treated as related, while in specific experiments, a distinction is made between proximal and distal knowledge. This inconsistency may affect the clarity and interpretability of the findings.
2. A second concern lies in the generalizability of the proposed approach. While the problem identified by the authors is indeed broadly relevant, the experiments are conducted on a relatively structured and constrained setting—primarily focusing on knowledge represented in the form of triples. However, in many real-world scenarios, knowledge does not naturally conform to such structured formats. It remains unclear how the proposed method would extend to more unstructured or complex forms of knowledge representation.

---

> ### Author Rebuttal · Authors · 2025-07-31
>
> We sincerely appreciate your kind and constructive feedback as well as your recognition of our work. Below are our responses to each of the concerns you raised.
>
> 1. **Clarification of related knowledge**: Thank you for your careful reading and thoughtful comments. Actually, the “related knowledge” we discuss in the paper is a gradable concept, which can be roughly categorized into three levels:
>
>    - **Level 1**: Knowledge within the medical domain that is semantically highly related to the injected knowledge (*proximal*);
>    - **Level 2**: Knowledge within the medical domain that is less semantically related (*distal*);
>    - **Level 3**: Knowledge outside the medical domain.
>
>    Our experimental findings reveal that the catastrophic forgetting caused by medical knowledge injection becomes progressively less severe from Level 1 to Level 3, exhibiting a **proximal-dependent** pattern. We sincerely apologize for not articulating this point clearly in the paper. We will revise the relevant definition in the updated version to clarify this concept.
>
>    1. A second concern lies in the generalizability of the proposed approach. While the problem identified by the authors is indeed broadly relevant, the experiments are conducted on a relatively structured and constrained setting—primarily focusing on knowledge represented in the form of triples. However, in many real-world scenarios, knowledge does not naturally conform to such structured formats. It remains unclear how the proposed method would extend to more unstructured or complex forms of knowledge representation.
>
> 2. **Generalizability issue**: We sincerely appreciate your insightful comments. In this work, we primarily focus on structured medical knowledge primarily because structured knowledge bases (e.g., DrugBank, UMLS) are abundant in the medical domain and represent one of the fundamental forms of medical knowledge. Moreover, the nature of structured knowledge allows for more precise measurement of semantic proximity, aiding the analysis of the forgetting pattern. That said, we agree that in real-world applications, knowledge also appears in unstructured forms such as free-text clinical narratives. Based on your constructive feedback, we conducted a preliminarily study to explore forgetting behaviors when injecting unstructured knowledge and to validate the effectiveness of our proposed method in mitigating such forgetting.
>
>    Specifically, we randomly sampled 2,000 clinical guidelines from a publicly available dataset [1], and used GPT-4.1 to generate 5 multiple-choice questions (MCQs) for each guideline. A subset of these questions was manually reviewed and found to be largely reliable for evaluation. We used these MCQs to evaluate the performance of LLaMA3-8B and selected 185 guidelines with accuracy below 50% as the injection knowledge set ($K_{\text{inject}}$), and 1,542 guidelines with accuracy above 75% as the evaluation set ($D_{\text{Eval}}$) to monitor forgetting. We adopt continued pretraining (**CPT**) as our approach for knowledge injection. Given the limited amount of injection data, we utilize commonly used data augmentation techniques [2,3], generating multiple paraphrased versions of the training samples in order to enhance the diversity of injection. Built on that, we further extend our proposed method (**InternAL**) to the unstructured knowledge. Specifically, we first extract key medical entities from each training sample using the target LLM, then prompt the model to recall relevant knowledge associated with these entities, and finally integrate the recalled knowledge into training samples to construct enriched pretraining texts. The evaluation results are summarized below:
>
> | Method                | $D_{\text{inject}}$ (Injected) | $D_{\text{eval}}$ (Uninjected) | MedQA   | MMLU-Med | MMLU-O  | ARC-C   | CSQA    |
> | --------------------- | -- | ---------------------------------------- | ------- | -------- | ------- | ------- | ------- |
> | Original              | 29.7  | 93.7  | 50.7    | 69.8     | 59.8    | 75.4    | 66.4    |
> | CPT                   | 49.6 | 85.9 | 44.3    | 63.6     | 55.8    | 72.1    | 61.0    |
> | *Relative Forgetting* | -| 8.3 | 12.6    | 8.9      | 6.7     | 4.4     | 8.1     |
> | **InternAL (ours)**   | 47.0 | 87.8 | 46.0    | 65.8     | 56.7    | 72.4    | 62.5    |
> | *Relative Forgetting* | -  | **6.3**  | **9.3** | **5.7**  | **5.2** | **4.0** | **5.9** |
>
> We observed the following phenomena from the results: (1) Continued Pretraining (CPT) achieves considerable performance on tasks related to the injected knowledge, but leads to significant forgetting, which exhibits proximity-dependent forgetting characteristics (a drop of 7.8 on $D_{\text{eval}}$, an average decrease of 6.3 on medical benchmarks, and an average decrease of 4.2 on general datasets); (3) Incorporating the internal relevant knowledge of LLMs into the training data (InternAL) can effectively mitigate forgetting, especially on the medical evaluation sets.
>
> We sincerely appreciate your suggestion and will include these additional experiments and analyses in the revised paper to discuss the generalizability of our findings and methods to unstructured knowledge.
>
> 3. **Reason for choosing RefInject**: Thank you for your thoughtful comment. We chose the RefInject method for knowledge injection for the following reasons: (1) Several prior studies in the medical domain [1,4,5] have demonstrated that training on MCQ datasets can effectively inject medical knowledge into LLMs and improve their performance on downstream medical tasks. (2) Other work has pointed out that continued pretraining on limited text data may fail to achieve effective generalization [6], whereas using corpora that contain implicit associations between entities (e.g., MCQ) has been shown to facilitate more efficient knowledge learning [2]. Motivated by these findings, we developed the RefInject method, which constructs MCQ-like questions that encode implicit associations between entities in the injected knowledge, aiming to enhance injection efficiency. Nevertheless, the proximity-dependent forgetting pattern we observed is **not** unique to this injection method. As mentioned in our previous response, we also observed this pattern when injecting knowledge through continued pretraining on unstructured clinical guidelines. We will further refine the relevant discussion in the revised manuscript to better clarify our motivation for choosing RefInject.
>
> 4. **Clarification of the experimental results**: We are very grateful for your thorough reading. The primary goal of knowledge injection is to supplement the missing knowledge in LLMs and improve their performance on tasks related to the injected knowledge. In this work, we randomly sampled 20,864 knowledge triples from PrimeKG and identified a subset where the LLMs performed poorly as the target knowledge for injection. After the injection, the model showed significant improvement on evaluation tasks corresponding to the injected knowledge (e.g., 9.7 to 77.4 for RefInject on $D_{\text{inject}}$). Although some forgetting occurred on the remaining triples, the overall performance across all 20,864 knowledge triples is improved by the injection (as shown in the $D_{\text{total}}$ column of the table below):
>
> | Model                | $D_{\text{inject}}$ | $D_{\text{eval}}$ | $D_{\text{total}}$ |
> | -------------------- | ----------------------------- | --------------------------- | ---------------------------- |
> | Original (Llama3-8B) | 9.7 | **91.4** | 51.5  |
> | RefInject            | **77.4** | 60.3 | 65.0 |
> | RefInject+GenFT      | 73.4  | 71.4 | 68.8  |
> | **InternAL (ours)**  | 74.3 | 70.9 | 69.3|
> | **InternAL+GenFT**   | 71.4 | 77.4 | **71.2**|
>
> The results above demonstrate that the baseline injection method (RefInject) improves performance on all knowledge triples by 13.5% compared to the original model. In contrast, our proposed method (RefInject) further alleviates forgetting and achieves an additional 4.3% improvement across all triples, while also effectively mitigating forgetting on external benchmarks.
>
> Thank you again for your kind comments. We will further clarify the relevant statements and add the total performance $D_{\text{total}}$ in the revised version to clarify the purpose and effectiveness of knowledge injection.
>
> 5. **Number Mismatch Issue**: Thank you for your careful reading. In fact, the results in lines 197–198 were from our initial single-run experiments. To verify robustness, we later conducted two additional independent runs and updated the table to report the average over all three runs, which showed consistent trends. However, we overlooked updating the corresponding description in the text. We apologize for the oversight and will correct this in the revised manuscript.
>
> 6. **Paper Formatting Concerns**: Thank you for your careful reading and helpful comment regarding Tables 2 and 3. We will address this in the revised version to improve clarity and presentation.
>
> [1] Chen Z, Cano A H, Romanou A, et al. Meditron-70b: Scaling medical pretraining for large language models. arXiv preprint arXiv:2311.16079, 2023.
>
> [2] Cabezudo M A S, Inácio M L, Pardo T A S. Investigating Paraphrase Generation as a Data Augmentation Strategy for Low-Resource AMR-to-Text Generation. Proceedings of the 17th International Natural Language Generation Conference. 2024.
>
> [3] Zhang X, Li M, Wu J. Co-occurrence is not factual association in language models. NeurIPS 2024.
>
> [4] Singhal, K., Tu, T., Gottweis, J. *et al.* Toward expert-level medical question answering with large language models. Nature Medicine, 2025.
>
> [5] Han T, Adams L C, Papaioannou J M, et al. MedAlpaca--an open-source collection of medical conversational AI models and training data[J]. arXiv preprint, 2023.
>
> [6] Christopher Potts, and Danqi Chen. Mquake: Assessing knowledge editing in language models via multi-hop questions. EMNLP 2023.

---

> > ### Comment · Reviewer_gLAz · 2025-08-06
> >
> > Thank you for your elaborated response. I am satisfied with your clarification and will consider raise my rating score. But I have some follow-up questions that are just for discussion purpose, your response to those questions won't change my opinion on this work anymore, but it will be good if you can enlight me more on this topic. So, from your further experiments on the unstructured knowledge, it feels like there is a trade of between applying CPT or InternalAL. CPT might achieve better performance on a specific dataset but will forget on other datasets. Your proposed method can avoding forgetting but may not be as competitive as the CPT method on the designated dataset. And applying your method involving extracted knowledge and curated extra dataset, this may also cost extra resources. IMO, training an expert model might be a more efficient way. But your proposed method may help those colossal models increase performance on ad-hoc scenarios without forgetting their versatile skills. It is interesting to observe and discuss this forgetting phenomen on and I think this work give us a solid verification on this forgetting mechanism. So, I recommond this work to be accepted by the conference, even the solution is not perfect.

---

> > > ### Author Response · Authors · 2025-08-06
> > >
> > > We sincerely thank you for the thoughtful review and insightful observations, which reflect a deep understanding of the challenges in this area. We greatly appreciate the opportunity you provided for further discussion, which has been highly inspiring for improving our paper and guiding our future research.
> > >
> > > 1. “…it feels like there is a trade of between applying CPT or InternAL”: This is indeed an interesting topic. When looking solely at performance metrics, the problem appears to be a “trade-off”. However, in real-world medical applications, even if newly injected knowledge is well acquired, forgetting related knowledge can render the model ineffective in practice. For example, a model may learn the new fact that “fever can be treated with ibuprofen,” but if it forgets that “ibuprofen should not be used in patients who are allergic to it,” then, despite knowing the right treatment, it may still recommend the wrong medication.
> > >
> > >    Moreover, in our experiments on unstructured knowledge, both test sets—$D_{\text{inject}}$ and $D_{\text{eval}}$—were derived from the same clinical guideline dataset. Specifically, $D_{\text{inject}}$ contains 925 questions evaluating the mastery of newly injected knowledge, covering 185 guidelines, while $D_{\text{eval}}$ consists of 7,710 questions aimed at assessing knowledge forgetting, spanning 1,542 guidelines. Although our method (InternAL) exhibits a 2.6% drop in accuracy on $D_{\text{inject}}$ compared to CPT, it achieves a 1.9% improvement on $D_{\text{eval}}$. When considering the entire guideline dataset, InternAL ultimately yields a 1.4% gain in micro-averaged accuracy over CPT, indicating an overall improvement in the model’s grasp of the guideline knowledge.
> > >
> > >    Notably, our proposed method (InternAL) may yield further improvements if more effectively adapted to unstructured data. At the same time, from the perspective of the no-free-lunch principle, a certain degree of trade-off between acquiring new knowledge and preserving existing knowledge may be unavoidable.
> > > 2. Additional computational cost: Indeed, as you rightly pointed out, our method involves model inference during internal knowledge extraction and data curation, resulting in additional computational cost. However, our method is designed to extract internal knowledge from the LLM related to a relatively small number of injected knowledge points, rather than performing large-scale inference computations. As a result, the additional computational cost is relatively modest. For example, in our experiments, extracting relevant internal knowledge for over 7,000 knowledge triples using llama3-8B took less than 40 minutes on a single A800 GPU, whereas large-scale continued pretraining typically requires hundreds of GPU hours. In addition, since our method primarily leverages the extracted knowledge to augment existing training examples—rather than introducing new data—it does not increase the size of the training set, and therefore incurs negligible additional cost during training.
> > >
> > >    More importantly, we consider this additional computation a reasonable trade-off given the benefits. A desirable learning process for LLMs should emulate that of humans—integrating new knowledge with existing internal knowledge, rather than focusing solely on the new, which can lead to forgetting what was previously learned.
> > >
> > > 3. Expert model or knowledge injection for colossal model: Your point is well taken — for highly specific tasks, training an expert LLM is indeed a reasonable and efficient approach. However, several studies [1,2] have shown that larger LLMs tend to possess stronger versatile capabilities, such as instruction following and reasoning, which are particularly valuable for addressing complex medical tasks. Nevertheless, these colossal models are reported to lack sufficient medical knowledge [3,4], highlighting the need to inject necessary medical knowledge — which is precisely the focus of our work. Of course, for highly specific tasks, expert models can still be a strong choice. Overall, this is an interesting question that depends heavily on the specific application scenario.
> > >
> > > [1] Kaplan J, McCandlish S, Henighan T, et al. Scaling laws for neural language models. arXiv preprint arXiv:2001.08361, 2020.
> > >
> > > [2] Wei J, Tay Y, Bommasani R, et al. Emergent Abilities of Large Language Models. Transactions on Machine Learning Research.
> > >
> > > [3] Hager, P., Jungmann, F., Holland, R. et al. Evaluation and mitigation of the limitations of large language models in clinical decision-making. Nature Medicine 30, 2613–2622, 2024.
> > >
> > > [4] Zuo Y, Qu S, Li Y, et al. MedXpertQA: Benchmarking Expert-Level Medical Reasoning and Understanding. ICML 2025.

---

### Official Review · Reviewer_w9Ko · 2025-07-06

**Clarity:** 3
**Significance:** 2
**Originality:** 2
**Rating:** 4
**Confidence:** 4

**Summary:**

This paper investigates catastrophic forgetting in LLMs when injecting medical knowledge via fine-tuning. The authors reveal a proximity-dependent pattern of forgetting: knowledge semantically close to the injected content is more susceptible to being overwritten, while unrelated knowledge is largely unaffected. The study empirically assesses several existing mitigation techniques and finds their performance lacking in the medical setting, particularly for knowledge close to injected facts. Inspired by mechanisms of human learning, the authors introduce InternAL, which augments the injection dataset by probing the LLM for its own closely related internal knowledge and incorporating these outputs into fine-tuning. Experiments with LLama3-8B and Qwen3-8B across a range of medical and general benchmarks demonstrate that InternAL significantly reduces proximity-based forgetting while maintaining injection effectiveness.

**Questions:**

Check the weakness part mentioned above.

**Ethical Concerns:**

["NO or VERY MINOR ethics concerns only"]

**Final Justification:**

I am satified with the author's response.

**Limitations:**

Check the weakness part mentioned above.

**Quality:**

2

**Strengths And Weaknesses:**

Strengths:
Well-Motivated Problem Formulation: The paper addresses an under-explored and practically pertinent issue: catastrophic forgetting during domain-specific knowledge injection in LLMs, with a focus on the medical domain.
Practical solution : InternAL is simple yet effective, leveraging the model’s own knowledge to mitigate forgetting.

Weaknesses:
Limited scope : Only tested in medical domain; other domains might behave differently.
Model size limitation : Experiments only cover 8B-parameter models (LLama, Qwen). Larger (e.g., 30B) or smaller (e.g., 1B) models may exhibit different forgetting dynamics, but this is untested.

---

> ### Author Rebuttal · Authors · 2025-07-31
>
> We sincerely appreciate your constructive feedback. Below, we address each of the concerns you raised.
>
> 1. **Research scope issue**: We sincerely appreciate your insightful comments. In fact, our focus on the medical domain is primarily motivated by a key observation: current LLMs still lack the necessary medical knowledge to perform reliably in real-world clinical scenarios. Our work investigates the phenomenon of catastrophic forgetting caused by medical knowledge injection and proposes an internal knowledge augmentation learning method to mitigate this issue, thereby improving the effectiveness of medical knowledge injection. Consequently, the majority of our experiments are conducted in the medical domain, which we also stated in the *Limitation* section.
>
>    Nevertheless, considering the underlying mechanisms of knowledge injection, the proximal-dependent forgetting pattern identified in this work may also exist in other domains. Based on your constructive suggestion, we conducted an additional small-scale study beyond the medical field. Specifically, we selected **human geography** as the target domain and extracted all **sister city relationships** from Wikidata (i.e., long-term partnerships between cities established through official agreements), sampling 20,000 city pairs for experimentation.
>
>    Following the same methodology used in our paper, we constructed evaluation questions to identify a subset of knowledge that was poorly mastered by the model (6,857 pairs selected for injection, denoted as $K_{\text{inject}}^{\text{SisCity}}$), and a well-mastered subset with model accuracy over 75% (4,145 pairs selected for evaluating forgetting, denoted as $D_{\text{eval}}^{\text{SisCity}}$). We then applied both the baseline method (**RefInject**) and our proposed method (**InternAL**) for knowledge injection. For evaluation, we leverage sister-city-based test sets $D_{\text{inject}}^{\text{SisCity}}$ and $D_{\text{eval}}^{\text{SisCity}}$ as well as on a suite of general-domain benchmarks. Furthermore, to evaluate the model’s forgetting of domain-related but semantically distant knowledge, we constructed an additional test set, *CityLoc*, by generating 7,174 questions based on the latitude and longitude information of cities extracted from Wikidata. We also studied the effect of general-domain finetuning (GenFT), an effective approach for mitigating catastrophic forgetting in the general domain. Experiments are conducted based on Llama3-8B, and the results are as follows:
>    |Model|$D_{\text{inject}}^{SisCity}$(Injected)|$D_{\text{eval}}^{SisCity}$ (Uninjected)|CityLoc|
>    |-|-|-|-|
>    |Original (Llama3-8B)|9.1|89.3|72.6|
>    |RefInject|93.9|64.3|67.2|
>    |RefInject+GenFT|93.0|69.0|66.8|
>    |**InternAL (ours)**|93.9|82.0|72.6|
>    |**InternAL**+GenFT|94.5|83.9|72.1|
>
>    The results above show that (1) direct knowledge injection (RefInject) leads to a 25% forgetting rate on $D_{\text{eval}}^{\text{SisCity}}$ and a 5.4% forgetting rate on CityLoc, while general-domain finetuning (GenFT)  fails to effectively address the substantial forgetting on $D_{\text{eval}}^{\text{SisCity}}$ and CityLoc; (2) Our method (InternAL) significantly mitigates forgetting on $D_{\text{eval}}^{\text{SisCity}}$ (from 64.3% to 82.0%) and on CityLoc (from 67.2% to 72.6%). Once again, thank you for your valuable feedback, and we will include this experiment in the revised version to support the generalizability of our main findings.
>
> 2. **Model size issue**: Thank you for your constructive suggestions. We selected the LLaMA and Qwen model families for our experiments primarily because they are representative open-source models with strong performance on several medical-domain benchmarks [1–2]. Considering the clinical applicability of LLMs, we primarily focus on the 8B model size, which offers a practical trade-off between computational resources, deployment cost, response latency, and reliability in real-world clinical environments. Larger models are often impractical to deploy due to their high resource demands and slower inference speed, while smaller models may lack sufficient medical knowledge to ensure reliable performance.
>
>    Indeed, models of different sizes may exhibit distinct forgetting dynamics. Nevertheless, our preliminary experiments also revealed similar forgetting patterns across different model sizes. Motivated by your constructive comment, we further conducted additional experiments on Qwen3-1.7B (full-parameter finetune) and Qwen3-32B (using LoRA due to limited computational resources). The results are as follows:
>
>    | Qwen3-1.7B          | $D_{\text{inject}}$ (Injected) | $D_{\text{eval}}$ (Uninjected) | MedQA | MMLU-Med | MMLU-O | ARC-C | CSQA |
>    | ------------------- | ---------------------------------------- | ---------------------------------------- | ----- | -------- | ------ | ----- | ---- |
>    | Original            | 9.7                                      | 88.7                                     | 37.5  | 59.0     | 52.5   | 71.6  | 66.4 |
>    | RefInject           | 61.3                                     | 66.2                                     | 29.2  | 48.2     | 46.0   | 59.9  | 53.9 |
>    | RefInject+GenFT     | 60.4                                     | 72.8                                     | 31.3  | 55.0     | 52.1   | 68.1  | 63.3 |
>    | **InternAL (ours)** | 59.3                                     | 75.1                                     | 31.9  | 51.7     | 47.4   | 61.7  | 58.3 |
>    | **InternAL**+GenFT  | 58.7                                     | 79.2                                     | 33.3  | 58.0     | 52.0   | 68.5  | 61.8 |
>
>    | Qwen3-32B           | $D_{\text{inject}}$ (Injected) | $D_{\text{eval}}$ (Uninjected) | MedQA | MMLU-Med | MMLU-O | ARC-C | CSQA |
>    | ------------------- | ---------------------------------------- | ---------------------------------------- | ----- | -------- | ------ | ----- | ---- |
>    | Original            | 10.2                                     | 92.4                                     | 68.2  | 80.8     | 68.5   | 86.9  | 83.5 |
>    | RefInject           | 71.1                                     | 62.7                                     | 59.1  | 77.6     | 67.6   | 84.7  | 84.4 |
>    | RefInject+GenFT     | 73.6                                     | 68.6                                     | 60.3  | 80.1     | 70.7   | 89.0  | 84.1 |
>    | **InternAL (ours)** | 64.7                                     | 72.8                                     | 62.5  | 79.6     | 67.9   | 86.7  | 82.7 |
>    | **InternAL**+GenFT  | 67.7                                     | 83.1                                     | 63.2  | 81.8     | 72.5   | 90.0  | 83.0 |
>
> Experimental findings are summarized as follows:
>
> 1. Both Qwen3-1.7B and Qwen3-32B exhibit **proximity-dependent forgetting**: For Qwen3-1.7B, knowledge injection (RefInject) led to performance degradation across multiple evaluation sets. While general-domain finetuning (GenFT) effectively mitigated forgetting in the general domain, the forgetting in the medical domain remained substantial. In contrast, Qwen3-32B showed relatively minor forgetting in the general domain after RefInject, but more pronounced forgetting in the medical domain (e.g., MedQA dropped from 68.2 to 59.1), which could not be effectively mitigated by GenFT.
> 2. Our proposed method, **InternAL**, effectively alleviates catastrophic forgetting in the medical domain for both models. Furthermore, it can be combined with GenFT to achieve even better mitigation of forgetting.
>
> We sincerely thank the reviewer for the insightful suggestion. We will include this experiment in the revised version of the paper to better support the observed forgetting patterns and the generalizability of our proposed method.
>
> [1] Wu C, Qiu P, Liu J, et al. Towards evaluating and building versatile large language models for medicine. npj Digital Medicine, 2025.
>
> [2] Zhu S, Hu W, Yang Z, Yan J, Zhang F. Qwen-2.5 Outperforms Other Large Language Models in the Chinese National Nursing Licensing Examination: Retrospective Cross-Sectional Comparative Study. JMIR Med Inform. 2025.

---

> > ### Author Response · Authors · 2025-08-05
> > **A brief summary of the rebuttal**
> >
> > Dear Reviewer w9Ko,
> >
> > Thank you for your time and feedback. This paper’s core contribution lies in identifying a previously underexplored phenomenon: LLMs tend to forget knowledge that is semantically related to newly injected knowledge in the medical domain. Based on this finding, we propose InternAL, a method that mitigates such forgetting by augmenting the injection input with relevant knowledge extracted from the target LLM. In response to your concerns, we have provided corresponding responses in the rebuttal, summarized as follows:
> >
> > + Research scope: Our research mainly focuses on the medical domain because current LLMs still lack enough medical knowledge for practical use. To test whether our findings generalize to other domains, we conducted a preliminary study in the human geography domain. Experimental results show that our method (InternAL) can also significantly reduce forgetting in this domain, supporting its effectiveness beyond medicine.
> > + Model size: We primarily focus on 8B models in our study for their practical balance of performance and efficiency in clinical scenarios. We further tested smaller (1.7B) and larger (32B) Qwen models and observed similar forgetting patterns across scales. Moreover, our method (InternAL) can effectively reduce such forgetting across diverse model sizes. These results will be added to the revised paper to demonstrate the method’s robustness across model sizes.
> >
> > Thank you again for reviewing our submission. We welcome any further questions or suggestions you may have.
> >
> > Best regards,
> >
> > Authors of Submission 26858

---

> ### Author Response · Authors · 2025-08-07
> **Confirmation on Addressing Your Concerns**
>
> Dear Reviewer w9Ko,
>
> Thank you again for your time and review.
>
> We noticed that you have clicked the “Mandatory Acknowledgement” button. We would like to kindly ask whether the additional experiments and clarifications provided in our rebuttal have sufficiently addressed your concerns. If you have any remaining questions, we would be very happy to engage in further discussion.
>
> Best regards,
>
> Authors of Submission 26858

---

### Note · Authors · 2025-08-12

We sincerely thank the area chair and all reviewers for their efforts in handling our submission and for this final opportunity to summarize our work. This paper reveals a previously underexplored proximity-dependent forgetting pattern in medical knowledge injection, where knowledge semantically closer to the injected facts is more likely to be forgotten. Furthermore, a method is introduced to mitigate such forgetting by fusing external injected knowledge with the LLM’s internal knowledge. This study provides insights into mechanisms and methods for continual knowledge injection into LLMs while avoiding forgetting. We are pleased to see that the reviewers recognized the core contributions of this work: (1) addressing a critical yet underexplored problem—catastrophic forgetting during domain-specific knowledge injection in LLMs (Reviewers w9Ko, gLAz, AipE); (2) providing a valuable analysis revealing the proximity-dependent forgetting pattern (Reviewers gLAz, ze9w); and (3) proposing the InternAL method, which is simple yet effective in mitigating such forgetting (all reviewers).

We understand that some reviewers may have had limited time to fully engage in the discussion, which could result in an incomplete view of our work. To facilitate a clearer understanding, we provide below a concise summary of the main concerns and our corresponding responses.

+ Regarding generalizability on unstructured data, we clarified that we focus on structured knowledge since it represents an important knowledge form in the medical domain. We further conducted a preliminary study, indicating that a similar forgetting pattern also exists on unstructured data and that our method can be generalized to such setting.

+ Regarding generalizability to other domains, we restated that our work focuses on the medical domain because current LLMs still lack sufficient medical knowledge for practical use. We also carried out a preliminary study in another domain, indicating that our method may generalize to other domains.

+ Regarding the effect of our method on the hallucination of LLMs, we clarified that the knowledge used for augmentation is extracted from the target model itself, and therefore does not further amplify the model's hallucination level. We further support our claim with a manual analysis on LLMs’ extracted knowledge.

We will further incorporate these clarifications and experiments into the revised paper. Thank you again for your understanding and continued support.

---

### Decision · Program_Chairs · 2025-09-17

**Decision:**

Accept (poster)

**Comment:**

This paper tackles the important and practical problem of catastrophic forgetting in LLMs during domain-specific knowledge injection, with a focus on the medical field.

The paper's primary contribution is twofold. First, it provides a solid analysis that identifies and empirically validates a "proximity-dependent" forgetting phenomenon, where knowledge semantically close to the newly injected content is more likely to be forgotten. Second, it proposes InternAL, a simple yet effective method that mitigates this specific type of forgetting. The core idea of leveraging the model's own internal knowledge to create a rehearsal dataset for augmentation is both novel and resource-efficient.

The experimental analysis is thorough, and the authors have done an excellent job addressing reviewer concerns during the rebuttal period by providing additional experiments that strengthen their claims regarding generalizability across different domains, model sizes, and data formats.

The findings presented here offer valuable insights into the mechanisms of knowledge retention in LLMs and open a promising avenue for future research. I encourage the authors to continue their work in this important direction. Therefore, I recommend this paper for acceptance.